# Unlabeled Data Can Provably Enhance In-Context Learning of Transformers

**Renpu Liu**
University of Virginia
Charlottesville, VA 22903
renpu@virginia.edu

**Jing Yang**
University of Virginia
Charlottesville, VA 22903
yangjing@virginia.edu

## Abstract

Large language models (LLMs) exhibit impressive in-context learning (ICL) capabilities, yet the quality of their predictions is fundamentally limited by the few costly *labeled* demonstrations that can fit into a prompt. Meanwhile, there exist vast and continuously growing amounts of *unlabeled* data that may be closely related to the ICL task. How to utilize such unlabeled data to provably enhance the performance of ICL thus becomes an emerging fundamental question. In this work, we propose a novel *augmented ICL* framework, in which the prompt includes a small set of labeled examples alongside a block of unlabeled inputs. We focus on the multi-class linear classification setting and demonstrate that, with chain-of-thought (CoT) prompting, a multi-layer transformer can effectively emulate an expectation-maximization (EM) algorithm. This enables the transformer to implicitly extract useful information from both labeled and unlabeled data, leading to provable improvements in ICL accuracy. Moreover, we show that such a transformer can be trained via teacher forcing, with its parameters converging to the desired solution at a linear rate. Experiments demonstrate that the augmented ICL framework consistently outperforms conventional few-shot ICL, providing empirical support for our theoretical findings. *To the best of our knowledge, this is the first theoretical study on the impact of unlabeled data on the ICL performance of transformers.*

## 1 Introduction

Since the introduction (Vaswani et al., 2017), transformers have become foundational models in diverse fields such as natural language processing (Radford, 2018; Devlin et al., 2019), computer vision (Dosovitskiy, 2020), and reinforcement learning (Chen et al., 2021). A key driver of their impact is the remarkable capability for In-Context Learning (ICL) (Brown et al., 2020). Without requiring parameter updates, transformers performing ICL can adapt to new tasks based solely on contextual examples provided within the prompt. This enables state-of-the-art few-shot performance across a multitude of applications, including reasoning and language understanding (Chowdhery et al., 2023), dialog generation (Thoppilan et al., 2022), and linear regression (Garg et al., 2022; Fu et al., 2023), etc.

Despite the power of ICL, its reliance on labeled examples presents a significant bottleneck for large language models (LLMs). Acquiring high-quality labeled data is in general expensive and time-consuming (Zhou et al., 2023; Chung et al., 2024; Sun et al., 2023; Wang et al., 2023). For example, creating the instruction-tuning and RLHF datasets for models like GPT-3.5 and GPT-4 involved thousands of expert annotator hours, yet constituted less than $0.1\%$ of the tokens encountered during pre-training (Ouyang et al., 2022; Achiam et al., 2023).

39th Conference on Neural Information Processing Systems (NeurIPS 2025).

Some existing approaches attempt to mitigate labeled data scarcity in ICL. For instance, Wan et al. (2023); Chen et al. (2025a) use an LLM to automatically generate pseudo-demonstrations at inference time by pairing unlabeled queries with the model's own predictions as pseudo labels. However, model-generated pseudo-labels inevitably inherit the biases and error patterns of the teacher model, resulting in noisy demonstrations that may limit potential performance gains.

In this work, instead of synthesizing examples with pseudo-labels, we explore a different approach by directly utilizing abundant and continuously growing (Raffel et al., 2020; Touvron et al., 2023) unlabeled data during ICL. The fundamental question we aim to answer is:

*Can we provably enhance the ICL performance of transformers by effectively leveraging plentiful unlabeled data alongside limited labeled examples?*

We answer this question affirmatively from a new *augmented in-context learning* perspective. This paradigm involves prompting a transformer with a mixture of a few labeled examples and numerous unlabeled examples, aiming to infer the missing labels within a single forward pass. By reasoning over unlabeled examples directly in the prompt, it bypasses the need for potentially costly, time-consuming, and bias-introducing labeling or pseudo-label generation steps in conventional ICL. In this work, we focus on augmented ICL for *multi-class linear classification*. Our main contributions are as follows.

- **Expressiveness with CoT Prompting.** First, we show that through Chain-of-Thought (CoT) prompting, a multi-layer transformer can leverage both labeled and unlabeled data to effectively solve the multi-class linear classification problem during ICL. Essentially, the transformer is able to obtain an initial estimation of the mean vectors of classes using the *labeled* data, and then iteratively refine the estimates by clustering the *unlabeled* data in an Expectation–Maximization (EM) fashion. We explicitly characterize the design of the transformer and theoretically prove that the class mean estimation will converge to the ground truth as the CoT steps increase. For a prompt consisting of $N$ labeled and $M$ unlabeled samples, the excess risk of our approach scales in $\mathcal{O}\left(1/\sqrt{N+\mathrm{poly}(M)}\right)$, strictly improving the excess risk lower bound of $\mathcal{O}\left(1/\sqrt{N}\right)$ for any classifier that utilizes $N$ labeled samples only. Our results indicate that the augmented ICL can effectively utilize the information from the unlabeled data, enabling steady performance improvement as unlabeled data increases.
- **Training Convergence under Teacher Forcing.** Second, we prove that, with proper initialization, when applying gradient descent on the population loss defined through teacher forcing, the tunable parameters of the transformer converge to the desired solution linearly. Thus, the trained transformer can mimic the EM algorithm through CoT prompting during inference, theoretically demonstrating that the expressive solution for augmented ICL is identifiable and learnable. Our proof involves a novel decomposition of the gradient of the CoT training loss into two analytically tractable terms. For each of them, we leverage the inherent isotropy of the involved quantities to simplify the analysis, which enables us to derive a tight upper bound on the critical inner-product term and obtain the linear convergence rate.
- **Empirical Results.** Finally, we evaluate the performance of augmented ICL in transformers trained via teacher forcing. Our experimental results show that the augmented ICL approach significantly outperforms conventional ICL in both class mean estimation and label prediction, with the advantage becoming more pronounced as the number of unlabeled data samples increases. Moreover, augmented ICL surpasses the Bayes-optimal classifier that relies solely on labeled data. These empirical observations are consistent with our theoretical findings.

## 2 Related Works

**ICL with Transformers.** Brown et al. (2020) first shows that GPT-3, a transformer-based LLM, can perform new tasks from input-output pairs without parameter updates, suggesting its ICL ability. This intriguing phenomenon of transformers has attracted much attention, leading to various interpretations and hypotheses about its underlying mechanism. Research on ICL often demonstrates how transformers can emulate learning algorithms. For instance, several studies have designed transformers that execute gradient descent for linear and non-linear regression tasks (Akyürek et al., 2023; Von Oswald et al., 2023a). Recent works demonstrate that transformers can implement more advanced optimization algorithms other than vanilla gradient descent on various ICL tasks (Bai et al.,

2024; Von Oswald et al., 2023b; Zhang et al., 2024a; Ahn et al., 2024; Liu et al., 2025). Another line of research adopts a statistical perspective: ICL can be viewed as an implicit form of Bayesian updating based on the examples provided in the prompt, with the diversity of pretraining data shaping the prior (Xie et al., 2022; Raventós et al., 2023; Garg et al., 2022).

Several studies (Gupta et al., 2024; Agarwal et al., 2024) investigate "unsupervised ICL", in which the prompt consists solely of unlabeled inputs. Another line of work leverages LLMs to generate pseudo-labels for unlabeled data, which are then used as demonstrations during ICL (Chen et al., 2023; Wan et al., 2023; Yang et al., 2023; Chen et al., 2025a). Our work leverages both labeled and unlabeled examples within the prompt to enhance ICL performance in a *semi-supervised learning* manner, which stands in sharp contrast to the aforementioned studies.

Notably, a recent concurrent work (Li et al., 2025) also investigates the impact of the semi-supervised data model on the ICL performance of transformers. Specifically, Li et al. (2025) focus on a linear transformer without nonlinear activations in a binary classification setting, and characterize the asymptotic ICL performance as the number of unlabeled samples approaches infinity. In contrast, we study a more realistic architecture that incorporates the softmax attention mechanism and establish a *non-asymptotic* convergence guarantee in the general *multi-class* setting.

**Training Dynamics of Transformers.** A number of recent works aim to provide theoretical characterizations of the training dynamics of transformers. Ahn et al. (2024); Mahankali et al. (2023); Zhang et al. (2024a); Huang et al. (2023) investigate the training dynamics of transformers with a single attention layer and a single head for in-context linear regression tasks. Cui et al. (2024) prove that transformers with multi-head attention layers outperform those with single-head attention. Cheng et al. (2024) show that local optimal solutions in transformers can perform gradient descent in-context for non-linear functions. Kim and Suzuki (2024) study the non-convex mean-field dynamics of transformers, and Nichani et al. (2024) characterize the convergence rate for the training loss in learning a causal graph. Additionally, Chen et al. (2024) investigate the gradient flow in training multi-head single-layer transformers for multi-task linear regression. Chen and Li (2025) propose a supervised training algorithm for multi-head transformers. The training dynamics of transformers for binary classification (Tarzanagh et al., 2023b,a; Vasudeva et al., 2024; Li et al., 2023; Deora et al., 2023; Li et al., 2024a), multi-class classification (Shen et al., 2025) and next-token prediction (Tian et al., 2023, 2024; Li et al., 2024b; Huang et al., 2024) have also been studied recently.

**Transformers with CoT.** In language modeling tasks, transformers have been proven to be powerful across various downstream tasks. However, transformers struggle to solve mathematical or scientific problems with a single generation, particularly when multiple reasoning steps are required. CoT prompting is introduced to enable transformers to generate intermediate results autoregressively before reaching the final answer, and has been shown to boost performance on arithmetic, commonsense, and scientific tasks (Wei et al., 2022; Kojima et al., 2022).

Recently, the training dynamics of transformers with CoT have been studied in Huang et al. (2025a) for weight prediction in linear regression, in Li et al. (2024a) for in-context supervised learning, in Kim and Suzuki (2025); Wen et al. (2025) for the parity problems, and in Huang et al. (2025b) for the even pairs problem. None of these studies, however, address whether the multi-step reasoning capacity through CoT can be utilized to extract information from *unlabeled* inputs.

# 3 Preliminaries

**Notations.** For matrix $\mathbf{X}$, we use $[\mathbf{X}]_{p:q,r:s}$ to denote the submatrix that contains rows $p$ to $q$ and columns $r$ to $s$, and we use $[\mathbf{X}]_{:,i}$ and $[\mathbf{X}]_{j,:}$ to denote the $i$-th column and $j$-th row of $\mathbf{X}$, respectively. For convenience, we occasionally denote the $i$-th column $\mathbf{X}$ by $[\mathbf{X}]_i$ when no ambiguity arises. $[\mathbf{X}]_{:,-C:-1}$ means the last $C$ columns of matrix $\mathbf{X}$. We use $\|\mathbf{X}\|_F$ to denote its Frobenius norm. For vector $\mathbf{x}$, we use $\|\mathbf{x}\|_1$, $\|\mathbf{x}\|$ and $\|\mathbf{x}\|_\infty$ to denote its $\ell_1$, $\ell_2$ and $\ell_\infty$ norms, respectively. We denote by $\mathbb{1}_d$ and $\mathbf{0}_d$ the $d$-dimensional all-1 and all-0 column vectors, respectively. $\mathbb{1}_{a \times b}$ and $\mathbf{0}_{a \times b}$ denote the all-1 and all-0 matrices of size $a \times b$, respectively. We denote the indicator function as $\mathbf{1}_{\{A\}}$, which equals 1 if event $A$ is true.

## 3.1 Transformer Architecture

In this work, we consider the encoder-based transformer architecture (Vaswani et al., 2017), where each transformer layer consists of an attention layer followed by a multi-layer perception (MLP) layer.

**Definition 3.1** (Attention layer). *Denote an $M$-head attention layer parameterized by $\{(\mathbf{V}_m, \mathbf{Q}_m, \mathbf{K}_m)_{m \in [M]}\}$ as $\mathrm{attn}_{\{(\mathbf{V}_m, \mathbf{Q}_m, \mathbf{K}_m)\}}(\cdot)$, where $\mathbf{V}_m, \mathbf{Q}_m, \mathbf{K}_m \in \mathbb{R}^{D \times D}$, $\forall m \in [M]$. Then, given an input sequence $\mathbf{H} \in \mathbb{R}^{D \times (N+1)}$, the output sequence of the attention layer is*

$$\mathrm{attn}_{\{(\mathbf{V}_m, \mathbf{Q}_m, \mathbf{K}_m)\}}(\mathbf{H}) = \mathbf{H} + \sum_{m=1}^{M} (\mathbf{V}_m \mathbf{H}) \times \sigma\big((\mathbf{K}_m \mathbf{H})^{\top}(\mathbf{Q}_m \mathbf{H})\big),$$

*where $\sigma$ is a non-linear activation function.*

**Definition 3.2** (MLP layer). *Given $\mathbf{W}_1 \in \mathbb{R}^{D' \times D}$, $\mathbf{W}_2 \in \mathbb{R}^{D \times D'}$ and a bias vector $\mathbf{b} \in \mathbb{R}^{D'}$, an MLP layer following the attention layer, denoted as $\mathrm{MLP}_{\{\mathbf{W}_1, \mathbf{W}_2, \mathbf{b}\}}$, maps each token in the input sequence (i.e, each column $\mathbf{h}_i$ in $\mathbf{H} \in \mathbb{R}^{D \times N}$) to another token as*

$$\mathrm{MLP}_{\{\mathbf{W}_1, \mathbf{W}_2, \mathbf{b}\}}(\mathbf{h}_i) = \mathbf{h}_i + \mathbf{W}_2 \sigma(\mathbf{W}_1 \mathbf{h}_i + \mathbf{b}),$$

*where $\sigma$ is a non-linear activation function.*

## 3.2 Augmented In-context Learning

**Conventional In-Context Learning (ICL).** For an ICL task, a trained transformer is given an ICL instance $\mathcal{I} = (\mathcal{D}, \mathbf{x}_{N+1})$, where $\mathcal{D} = \{(\mathbf{x}_j, y_j)\}_{j \in [N]}$ and $\mathbf{x}_{N+1}$ is a query. Here, $\mathbf{x}_j \in \mathbb{R}^d$ is an in-context example, and $y_j$ is the corresponding label for $\mathbf{x}_j$. For each instance, $\{(\mathbf{x}_j, y_j)\}_{j=1}^{N+1}$ are generated independently accordingly to an underlying distribution. The objective of ICL is to predict $y_{N+1}$ without any parameter updating of the transformer.

**Augmented ICL.** In this work, we consider a new unlabeled data augmented ICL framework. Specifically, each ICL instance now comprises a set of labeled examples, $\mathcal{D}_{\mathrm{label}} := \{(\mathbf{x}_j, y_j)\}_{j=1}^{N}$, and a set of unlabeled examples, $\mathcal{D}_{\mathrm{unlabel}} := \{\mathbf{x}_j\}_{j=N+1}^{N+M}$, i.e., $\mathcal{I} = \mathcal{D}_{\mathrm{label}} \cup \mathcal{D}_{\mathrm{unlabel}}$. Similar to conventional ICL, all $(\mathbf{x}_j, y_j)$ pairs follow the same distribution. The objective of augmented ICL is then to predict labels for all the $M$ unlabeled samples in $\mathcal{D}_{\mathrm{unlabel}}$.

We note that the augmented ICL generalizes the conventional ICL framework, and reduces to it when $M = 1$. While the conventional ICL can be utilized to solve the prediction for those $M$ unlabeled samples individually *in parallel*, by augmenting them in the same ICL instance, it provides an opportunity for the transformer to extract common statistical information in those unlabeled data, which can be utilized to improve the joint prediction accuracy.

**Augmented ICL for Multi-class Linear Classification.** We consider augmented ICL for a multi-class linear classification problem. We assume there exist $C$ classes, and the label space $\mathcal{Y}$ consists of one-hot vectors $\{\mathbf{e}_1, \dots, \mathbf{e}_C\}$, where each $\mathbf{e}_i \in \mathbb{R}^C$ is the $i$-th unit vector. For each ICL instance $\mathcal{I}_{\mathbf{M}}$, the samples are randomly generated according to

$$\mathbf{M} \sim P_{\mathbf{M}}, \quad \mathbf{y}_j \sim \mathrm{Uniform}(\mathcal{Y}), \quad \boldsymbol{\epsilon}_j \sim \mathcal{N}(\mathbf{0}, \boldsymbol{\Sigma}), \quad \mathbf{x}_j = \mathbf{M}\mathbf{y}_j + \boldsymbol{\epsilon}_j, \quad j \in [M+N], \quad (3.1)$$

where $\mathbf{M} \in \mathbb{R}^{d \times C}$ and $P_{\mathbf{M}}$ is a prior distribution over $\mathbb{R}^{d \times C}$. Denote the columns of $\mathbf{M}$ as $\{\boldsymbol{\mu}_i\}_{i=1}^{C}$. Then, each $\mathbf{x}_j$ essentially follows a $C$-component mixture Gaussian distribution parametrized by mean vectors $\{\boldsymbol{\mu}_i\}_{i=1}^{C}$ and shared covariance matrix $\boldsymbol{\Sigma}$. In this work, we assume $\boldsymbol{\Sigma}$ is isotropic. We adopt this assumption for theoretical tractability, as it is crucial for deriving the closed-form update rules for the transformer. This approach is a standard and widely adopted practice in related literature to facilitate theoretical analysis (He et al., 2025; Zhang et al., 2024b; Chen et al., 2025b).

## 3.3 Chain-of-Thought Prompting for Augmented ICL

The core challenge in augmented ICL is leveraging both unlabeled data and labeled examples to infer task structure from a single instance. Unlike standard few-shot ICL, which often uses direct pattern matching, the augmented ICL requires more complex inference to effectively utilize the larger unlabeled set, making a simple one-step prediction insufficient.

Chain-of-Thought (CoT) reasoning offers a promising way to enhance a transformer's ICL capabilities. This is crucial for augmented ICL, as it enables the transformer to effectively utilize unlabeled data through iterative latent parameter estimation and refinement.

To implement augmented in-context learning via CoT prompting, we first encode a task instance $\mathcal{I}$ into an embedding matrix $\mathbf{H}$ by concatenating three column blocks: the labeled example block, the unlabeled example block, and the reasoning block as follows:

$$\mathbf{H} = \left[ \begin{array}{ccccccccc} \mathbf{x}_1 & \cdots & \mathbf{x}_N & \mathbf{x}_{N+1} & \cdots & \mathbf{x}_{N+M} & \mathbf{0} & \cdots & \mathbf{0} \\ \mathbf{y}_1 & \cdots & \mathbf{y}_N & \mathbf{0} & \cdots & \mathbf{0} & \mathbf{0} & \cdots & \mathbf{0} \\ \mathbf{p}_1 & \cdots & \mathbf{p}_N & \mathbf{p}_{N+1} & \cdots & \mathbf{p}_{N+M} & \mathbf{q}_1 & \cdots & \mathbf{q}_C \end{array} \right] \triangleq \left[ \begin{array}{ccc} \mathbf{X}_\ell & \mathbf{X}_u & \mathbf{0} \\ \mathbf{Y}_\ell & \mathbf{0} & \mathbf{0} \\ \mathbf{P}_\ell & \mathbf{P}_u & \mathbf{Q}^{(0)} \end{array} \right], \quad (3.2)$$

where $\mathbf{p}_j \in \mathbb{R}^{d_p}$ is an auxiliary embedding that stores the (predicted) classification probability vector for the $j$-th sample, as well as a binary indicator to distinguish the labeled and unlabeled data. $\mathbf{q}_i \in \mathbb{R}^{d_p}$ serves as the initial CoT token for class $i$, which contains the one-hot vector $\mathbf{e}_i$ to indicate the corresponding class, and an all-zero vector representing the transformer's initial estimate for the mean vector $\boldsymbol{\mu}_i$.

Denote a trained transformer with parameter $\boldsymbol{\Theta}$ as $\mathrm{TF}_{\boldsymbol{\Theta}}$. With CoT, we will use the transformer to generate $T$ intermediate steps before it outputs the prediction. Specifically, let $\widehat{\mathbf{H}}^{(t-1)}$ be the input sequence at the $t$-th step of CoT, where $\widehat{\mathbf{H}}^{(0)} = \mathbf{H}$, and $\mathrm{TF}_{\boldsymbol{\Theta}}(\widehat{\mathbf{H}}^{(t-1)})$ as the corresponding output of the transformer. Then, we will take out the last $C$ columns of $\mathrm{TF}_{\boldsymbol{\Theta}}(\widehat{\mathbf{H}}^{(t-1)})$, and append them to the end of $\widehat{\mathbf{H}}^{(t-1)}$ to form the input for the next CoT step. Specifically, we have

$$\widehat{\mathbf{H}}^{(t)} = \left[ \widehat{\mathbf{H}}^{(t-1)}, [\mathrm{TF}_{\boldsymbol{\Theta}}(\widehat{\mathbf{H}}^{(t-1)})]_{:,-C:-1} \right] = \left[ \begin{array}{cccccc} \mathbf{X}_\ell & \mathbf{X}_u & \mathbf{0} & \star & \cdots & \star \\ \mathbf{Y}_\ell & \mathbf{0} & \mathbf{0} & \star & \cdots & \star \\ \mathbf{P}_\ell & \mathbf{P}_u & \mathbf{Q}^{(0)} & \mathbf{Q}^{(1)} & \cdots & \mathbf{Q}^{(t)} \end{array} \right], \quad (3.3)$$

where

$$\mathbf{Q}^{(t)} = \left[ \begin{array}{ccc} \mathbf{e}_1 & \cdots & \mathbf{e}_C \\ \widehat{\boldsymbol{\mu}}_1^{(t)} & \cdots & \widehat{\boldsymbol{\mu}}_C^{(t)} \\ \star & \cdots & \star \end{array} \right]. \quad (3.4)$$

Here $\star$ is a placeholder for dummy tokens, $\mathbf{e}_i$ is the $i$-th unit vector, and $\widehat{\boldsymbol{\mu}}_i^{(t)}$ is the estimated mean vector for class $i$ at the $t$-th CoT step.

After $T$ iterations, we read out $\widehat{\boldsymbol{\mu}}_1^{(T)} \cdots \widehat{\boldsymbol{\mu}}_C^{(T)}$ from $\mathbf{Q}^{(T)}$ as the final estimation of the class mean vectors. Then, the label of each unlabeled data can be estimated through a maximum likelihood estimation, i.e.,

$$\widehat{\mathbf{y}}_j = \left\{ \mathbf{e}_i : i = \arg \min_{i \in [C]} \left\| \mathbf{x}_j - \widehat{\boldsymbol{\mu}}_i^{(T)} \right\| \right\}, \quad j \in [N+1 : N+M]. \quad (3.5)$$

## 4 Expressiveness with CoT Prompting for Augmented ICL

In this section, we show that a multi-layer transformer *can* implement an Expectation-Maximization (EM)-style algorithm to extract useful statistical information from the unlabeled data, which will be combined with information extracted from the labeled data to jointly estimate the class means and improve the augmented ICL performance. Specifically, we have the following result.

**Theorem 4.1.** *There exists a 4-layer transformer, such that its output sequence at the $(t+1)$-th CoT step satisfies*

$$\widehat{\boldsymbol{\mu}}_i^{(t+1)} = \widehat{\boldsymbol{\mu}}_i^{(t)} - \frac{\eta^{(t)}}{M} \sum_{j=N+1}^{N+M} p_{ij}^{(t)} \left( \widehat{\boldsymbol{\mu}}_i^{(t)} - \mathbf{x}_j \right) + \mathbf{1}_{\{t=0\}} \cdot \frac{C}{N} \sum_{j=1}^{N} (\mathbf{e}_i^\top \mathbf{y}_j) \mathbf{x}_j, \quad (4.1)$$

*for any $i \in [C]$, where $\eta^{(t)} = \alpha/(T'+t)$ for some positive constants $\alpha$ and $T'$, $p_{ij}^{(t)}$ is the normalized weight*

$$p_{ij}^{(t)} = \frac{\sum_{\tau=0}^{t} \exp\left( -\frac{1}{2} \|\widehat{\boldsymbol{\mu}}_i^{(\tau)} - \mathbf{x}_j\|_{\boldsymbol{\Sigma}^{-1}}^2 + \beta\tau \right)}{\sum_{\tau=0}^{t} \sum_{c=1}^{C} \exp\left( -\frac{1}{2} \|\widehat{\boldsymbol{\mu}}_c^{(\tau)} - \mathbf{x}_j\|_{\boldsymbol{\Sigma}^{-1}}^2 + \beta\tau \right)}, \quad (4.2)$$

*and $\beta$ is a positive constant.*

We outline the construction of each layer of the transformer below, and defer the detailed derivation and specific parameter implementation to Appendix C.

The four-layer architecture is designed to mirror an EM iteration for Gaussian mixture model clustering (Zhao et al., 2020; Sula and Zheng, 2022) within the transformer's forward pass. The EM algorithm operates iteratively. First, in the **E-step**, it utilizes the current class mean estimates embedded in the input sequence to compute the estimated class membership for each *unlabeled* data point. Subsequently, the **M-step** updates the class mean estimates by performing a maximum likelihood estimation of the unlabeled data, and then combining them with the estimates obtained from the *labeled* data. Through this iterative process, the algorithm converges to accurate estimates of the underlying class means, enabling reliable classification.

**The first layer.** The first transformer layer includes a *softmax-activated* attention layer followed by an MLP layer. We construct its parameters so that it outputs the class membership estimate for the each unlabeled sample as in the form of Equation (4.2), where the mean estimates $\{\widehat{\boldsymbol{\mu}}_1^{(\tau)}, \cdots, \widehat{\boldsymbol{\mu}}_C^{(\tau)}\}_{\tau=1}^t$ are embedded in the reasoning blocks $\mathbf{Q}^{(1)} \cdots \mathbf{Q}^{(t)}$ in the input sequence, and the parameter $\beta$ is embedded in the first layer as well. This probability represents how likely sample $j$ is estimated to be in class $i$. Since the temperature parameter $\beta\tau$ is proportional to the step index $\tau$, estimates from earlier CoT steps carry less importance. In the limit of $\beta \to \infty$, the weight vector depends only on the latest CoT step, i.e.,

$$p_{ij}^{(t)} = \frac{\exp\left(-\frac{1}{2}\|\widehat{\boldsymbol{\mu}}_i^{(t)} - \mathbf{x}_j\|_{\boldsymbol{\Sigma}^{-1}}^2\right)}{\sum_{c=1}^C \exp\left(-\frac{1}{2}\|\boldsymbol{\mu}_c^{(t)} - \mathbf{x}_j\|_{\boldsymbol{\Sigma}^{-1}}^2\right)}. \tag{4.3}$$

**The second and third layers.** The second and third transformer layers consist of a *linear* attention layer followed by an MLP layer. These layers are designated for the M-step of the EM algorithm. In this step, the class mean estimates $\{\widehat{\boldsymbol{\mu}}_i\}_{i=1}^C$ are updated by maximizing the overall log-likelihood of the unlabeled data with the estimated class membership probabilities $p_{ij}^{(t)}$. It aims to solve

$$P_1: \quad \{\boldsymbol{\mu}_c^{(t+1)}\}_c = \arg\max_{\{\mu_c\}_c} \sum_{j=N+1}^{N+M} \sum_{i=1}^C p_{ij}^{(t)} \log \mathcal{N}(\mathbf{x}_j; \boldsymbol{\mu}_i, \boldsymbol{\Sigma}).$$

The implementation for these two layers is equivalent to tasking one step of gradient descent over $P_1$, i.e.,

$$\widehat{\boldsymbol{\mu}}_i^{(t+1)} = \widehat{\boldsymbol{\mu}}_i^{(t)} - \frac{\eta^{(t)}}{M} \sum_{j=N+1}^{N+M} p_{ij}^{(t)} \left(\widehat{\boldsymbol{\mu}}_i^{(t)} - \mathbf{x}_j\right). \tag{4.4}$$

**The fourth layer.** Finally, the last transformer layer includes a *ReLU-activated* attention layer followed by an MLP layer. This layer calculates the initial class mean estimates for the *labeled* dataset and is only activated at the first CoT step. It implements the following updating rule:

$$\widehat{\boldsymbol{\mu}}_i^{(t+1)} = \mathbf{1}_{\{t=0\}} \cdot \frac{C}{N} \sum_{j=1}^N (\mathbf{e}_i^\top \mathbf{y}_j) \mathbf{x}_j, \tag{4.5}$$

which initialize $\widehat{\boldsymbol{\mu}}_i^1$ to be the average of $\mathbf{x}_j$s for the labeled data samples in class $i$. This initialization will be refined iteratively through the CoT steps by leveraging the information from the unlabeled data.

We note that the parameters of the last three layers are data-independent and can be explicitly constructed beforehand, and only the parameters of the first layer depend on the distribution of the data, which can be obtained through CoT training, as elaborated in Section 5.

Next, we will show that the transformer specified in Theorem 4.1 will recover $\{\mu_i\}_{i=1}^C$ accurately with high probability, and explicitly characterize the benefit of unlabeled data in this augmented ICL.

**Theorem 4.2** (Class Mean Estimation Error). *Given the transformer described in Theorem 4.1, when $N \geq \frac{36\alpha^2 L^2}{c_1} \log 1/\epsilon$ and $M \geq \max\{36\alpha^2 L^2 K, \log^2(1/\epsilon)\}$, and $t \geq \max\{\sqrt[4]{M}, T'\}$, with*

*probability at least $1 - \epsilon$, the output of the transformer after $t$ CoT steps satisfies*

$$\|\widehat{\mathbf{M}}^{(t)} - \mathbf{M}\|_F^2 \leq c \frac{\log(1/\epsilon)}{N + \sqrt[4]{M}},$$

*where $c_1, c, \alpha, L, T', K$ are positive constants.*

*Proof sketch.* The proof of Theorem 4.2 contains three major steps. In **Step 1**, we utilize the Hoeffding's inequality to ensure that with a sufficient number of labeled data $N$, the initial class mean estimates $\widehat{\boldsymbol{\mu}}_1^{(1)}, \cdots, \widehat{\boldsymbol{\mu}}_C^{(1)}$ are in a small neighborhood of the ground truth class means $\boldsymbol{\mu}_1, \cdots, \boldsymbol{\mu}_C$. In **Step 2**, we need to bound the gap between the gradient descent updating step for $t > 1$ in Equation (4.4), and one gradient descent step for the expected log-likelihood loss $\mathcal{L}(\{\widehat{\boldsymbol{\mu}}_i^{(t)}\}) = \mathbb{E}_{\mathbf{x}}\left[\log\left(\frac{1}{C}\sum_{i=1}^C \exp\left(-\frac{1}{2}\|\mathbf{x} - \widehat{\boldsymbol{\mu}}_i^{(t)}\|^2\right)\right)\right]$. To ensure that the gap is sufficiently small, we need to design the temperature parameter $\beta\tau$ so that the normalized weight is biased heavily toward the class mean estimation obtained from the current CoT step, and the influence of previous CoT steps is minimized. Then, utilizing Bernstein's inequality, this gap is bounded. In **Step 3**, we utilize Lipschitz continuity of $\mathcal{L}(\{\boldsymbol{\mu}_i\})$, combing the bound on the gradient gap in Step 2, to show that $\|\widehat{\mathbf{M}}^{(t)} - \mathbf{M}\|_F^2 \leq \mathcal{O}(1/\sqrt{N + \text{poly}(M)})$ for $t$ large enough if $\widehat{\mathbf{M}}^{(1)}$ is in a small neighborhood of $\mathbf{M}$, which is guaranteed in Step 1. The complete proof can be found in Appendix C. $\qquad\square$

Based on the smoothness of the Bayes risk, we have the following corollary as a direct consequence of Theorem 4.2.

**Corollary 4.1** (Label Prediction Error Bound). *Let $\widehat{\mathbf{y}}_j$ be the predicted label for $\mathbf{x}_j$ according to Equation (3.5). Let $\mathcal{R}^*$ be the prediction error under the Bayes-optimal classifier with known class mean vectors $\boldsymbol{\mu}_1, \cdots, \boldsymbol{\mu}_C$. Then, under the same conditions as described in Theorem 4.2, we have*

$$\mathbb{P}[\widehat{\mathbf{y}}_j \neq \mathbf{y}|\boldsymbol{\mu}_1, \cdots, \boldsymbol{\mu}_C] - \mathcal{R}^* \leq \mathcal{O}\left(\frac{1}{\sqrt{N + \text{poly}(M)}}\right).$$

**Remark 1.** *The advantage of utilizing unlabeled data in the augmented ICL becomes evident when comparing Corollary 4.1 with the existing lower bound on the excess risk for classical binary classification. It has been shown that the excess risk for any classifier trained on $N$ labeled data scales in $\Omega(1/\sqrt{N})$ in the worst case of $\mathbf{M}$ (Li et al., 2017), which is in stark contrast to the upper bound $\mathcal{O}\left(1/\sqrt{N + \text{poly}(M)}\right)$ in Corollary 4.1. This result indicates that the designed transformer can effectively utilize the unlabeled data through CoT prompting, and strictly improves the prediction accuracy of any classifier that utilizes the labeled data only.*

## 5 Training Dynamics with Teacher Forcing

While Section 4 indicates that there exists a transformer that is able to implement an EM-type algorithm to utilize unlabeled data and improve the ICL performance through CoT prompting, in this section, we show that such a transformer can be obtained through teacher forcing training (Kim and Suzuki, 2025; Huang et al., 2025b).

The training objective of teacher forcing is to ensure that the transformer can mimic the trajectory of iterative updating under an EM algorithm during the CoT inference. Formally, we require that, on the unlabeled set, the cross-entropy between the class distributions induced by the CoT estimates $\{\widehat{\boldsymbol{\mu}}_c^{(t)}\}_{c=1}^C$ and those induced by the reference method $f_{\text{ref}}$ remains small for all $t = 1, \ldots, T$. Specifically, given $\mathbf{X}_\ell, \mathbf{Y}_\ell$ and $\mathbf{X}_u$, we denote the generated reference trajectory as $f_{\text{ref}}(\mathbf{X}_\ell, \mathbf{Y}_\ell, \mathbf{X}_u) = \{\boldsymbol{\mu}_{\text{ref},1}^{(t)} \cdots \boldsymbol{\mu}_{\text{ref},C}^{(t)}\}_{t=1}^T$. Then, we construct the reference embedding sequence at the $t$-th CoT step as

$$\mathbf{H}_{\text{ref}}^{(t)} = \begin{bmatrix} \mathbf{X}_\ell & \mathbf{X}_u & \mathbf{0} & \star & \cdots & \star \\ \mathbf{Y}_\ell & \mathbf{0} & \mathbf{0} & \star & \cdots & \star \\ \mathbf{P}_\ell & \mathbf{P}_u & \mathbf{Q}^{(0)} & \mathbf{Q}_{\text{ref}}^{(1)} & \cdots & \mathbf{Q}_{\text{ref}}^{(t)} \end{bmatrix}, \quad \mathbf{Q}_{\text{ref}}^\tau = \begin{bmatrix} \mathbf{e}_1 & \cdots & \mathbf{e}_C \\ \boldsymbol{\mu}_{\text{ref},1}^{(\tau)} & \cdots & \boldsymbol{\mu}_{\text{ref},C}^{(\tau)} \\ * & \cdots & * \end{bmatrix}, \forall \tau \in [t].$$

We note that the reference embedding shares the same structure as the embedding defined in Equation (3.3), except that now the mean estimates are generated by the reference algorithm instead of

the transformer itself. We then feed $\mathbf{H}_{\text{ref}}^{(t)}$ to the transformer, and extract the updated mean estimates from its output $\text{TF}_{\boldsymbol{\Theta}}(\mathbf{H}_{\text{ref}}^{(t)})$.

The corresponding CoT training loss can be defined as:

$$\widehat{\mathcal{L}}_{\text{CoT-train}}(\boldsymbol{\Theta}; \mathcal{I}_{\mathbf{M}}) = \frac{1}{T} \sum_{t=1}^{T} \sum_{j=N+1}^{N+M} \ell_{\text{CE}}\left(\mathbf{q}_j^{(t)}, [\text{TF}_{\boldsymbol{\Theta}}(\mathbf{H}_{\text{ref}}^{(t-1)})]_{2d+2c+1:2d+3c, N+j}\right), \qquad (5.1)$$

where $\ell_{\text{CE}}$ is the cross-entropy loss and $\mathbf{q}_j^{(t)} = [p_{1j}^{(t)} \cdots p_{Cj}^{(t)}]$ with

$$p_{ij}^{(t)} = \frac{\exp\left(-\frac{1}{2}\|\boldsymbol{\mu}_{\text{ref},i}^{(t)} - \mathbf{x}_j\|_{\Sigma^{-1}}^2\right)}{\sum_{c=1}^{C} \exp\left(-\frac{1}{2}\|\boldsymbol{\mu}_{\text{ref},c}^{(t)} - \mathbf{x}_j\|_{\Sigma^{-1}}^2\right)}.$$

Similar to Ahn et al. (2024); Huang et al. (2025a), in this work, we analyze the training convergence of the population loss defined as:

$$\mathcal{L}_{\text{CoT-train}}(\boldsymbol{\Theta}) = \mathbb{E}_{\mathcal{I}_{\mathbf{M}}}\left[\widehat{\mathcal{L}}_{\text{CoT-train}}(\boldsymbol{\Theta}; \mathcal{I}_{\mathbf{M}})\right], \qquad (5.2)$$

where the expectation is taken over the randomness in the generation process of $\mathcal{I}_{\mathbf{M}}$.

Directly analyzing the training dynamics of all layers of the transformer is intractable. On the other hand, as we mentioned in Section 4, the last three layers of the transformer can be constructed explicitly beforehand, as their parameters are data-independent. As a result, in the following, we will freeze these three layers and train the first layer only.

**Assumption 1** (Initialization). *We initialize the first layer of the three-layer transformer described in Theorem 4.1 as follows:*

$$\mathbf{Q}^{(0)}\mathbf{K}^{(0)} = \begin{bmatrix} \mathbf{0}_{d \times (d+2C)} & \mathbf{W}^{(0)} & & & \\ & & \mathbf{0}_{(4C+d+2) \times 2C} & & \\ & & & 1 & \mathbf{0}_{1 \times 2} & \beta^{(0)} \\ & & & & & 0 \end{bmatrix},$$

$$\mathbf{V}^{(0)} = \text{diag}\left(\mathbf{0}_{(d+2C) \times (d+C)}, \mathbf{I}_C, \mathbf{0}_{(d+C+4) \times (d+2C+4)}\right),$$

*where $\mathbf{W}^{(0)}$ is a $d \times d$ matrix whose entries are randomly sampled from a standard Gaussian distribution, $\beta^{(0)}$ is a constant, and all the unspecified entries are equal to zero.*

**Theorem 5.1** (Training Convergence). *Let $\{\mathbf{Q}^{(k)}, \mathbf{K}^{(k)}, \mathbf{V}^{(k)}\}_{k \geq 0}$ be the parameters of the first attention layer of the transformer after applying $k$ iterations of gradient descent on the population loss defined in Equation (5.2). Then, with the initialization specified in Assumption 1, we have*

$$\|\mathbf{W}^{(k)} - \boldsymbol{\Sigma}^{-1}\|_F^2 \leq c^k \|\mathbf{W}^{(0)} - \boldsymbol{\Sigma}^{-1}\|_F^2$$

*for some positive constant $c$, while the other parameters in $\mathbf{Q}^{(0)}$, $\mathbf{K}^{(0)}$ and $\mathbf{V}^{(0)}$ remain unchanged.*

Theorem 5.1 indicates that under teacher forcing training, the parameter matrix $\mathbf{W}^{(k)}$ of the first layer converges to $\boldsymbol{\Sigma}^{-1}$, the inverse of the noise covariance matrix, linearly. Combining with other parameters in $\mathbf{Q}^{(0)}$, $\mathbf{K}^{(0)}$ and $\mathbf{V}^{(0)}$, we observe that the teacher forcing training recovers the transformer described in Theorem 4.1, theoretically demonstrating that the expressive solution for augmented ICL is identifiable and learnable.

*Proof sketch.* We use the superscript $(k, t)$ to denote the $t$-th CoT step in the $k$-th gradient descent iteration. First, we drop the temperature term $\beta\tau$ in the definition of $p_{ij}^{(k,t)}$ given in Equation (4.2) and approximate it as in Equation (4.3), noting the approximation error can be made arbitrarily small by taking $\beta$ sufficiently large. Next, we define

$$q_{ij}^{(k,t)} = \frac{\exp\left(-\frac{1}{2}\|\widehat{\boldsymbol{\mu}}_i^{(k,t)} - \mathbf{x}_j\|_{\mathbf{W}^{(k)}}^2\right)}{\sum_{h=1}^{C} \exp\left(-\frac{1}{2}\|\widehat{\boldsymbol{\mu}}_h^{(k,t)} - \mathbf{x}_j\|_{\mathbf{W}^{(k)}}^2\right)},$$

which corresponds to replacing $\boldsymbol{\Sigma}^{-1}$ by $\mathbf{W}^{(k)}$ in the approximation of $p_{ij}^{(k,t)}$.

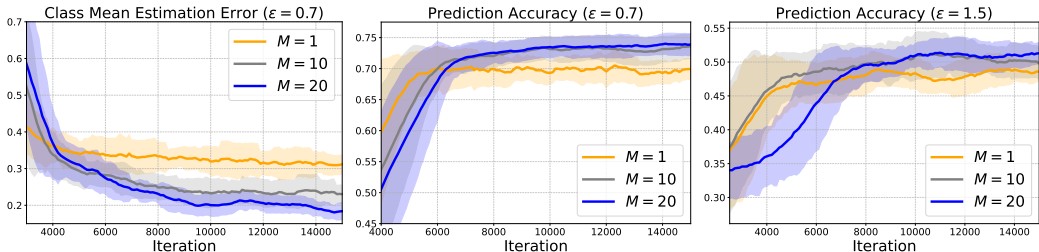

Figure 1: Inference performance of the transformer trained via teacher forcing versus number of gradient descent iterations during training. Number of classes $C = 3$, number of labeled examples $N = 5$, CoT steps $T = 5$. The solid line shows the average results across 5 runs, and the shaded region represents $\pm 2$ standard deviations.

To prove one-step improvement of gradient descent on the population loss under teacher forcing, we must exhibit a constant $\alpha > 0$ such that $-\langle \mathbf{W}^{(k)} - \mathbf{\Sigma}^{-1}, \eta^{(k)} \nabla_{\mathbf{W}} \mathcal{L}_{\mathrm{CoT-train}} \rangle \leq -\alpha \|\mathbf{W}^{(k)} - \mathbf{\Sigma}^{-1}\|_F^2$. Our proof proceeds in three major steps. **Step 1.** Since direct analysis of $\nabla_{\mathbf{W}} \mathcal{L}$ is intractable, we propose a novel decomposition by applying Stein's lemma to break the gradient into two analytically tractable terms: one is the posterior-difference term involving $\mathbb{E}[\mathbf{p}_j^{(k,t)} - \mathbf{q}_j^{(k,t)}]$ and the other is the Jacobian-difference term involving $\mathbb{E}[\nabla \mathbf{p}_j^{(k,t)} - \nabla \mathbf{q}_j^{(k,t)}]$. **Step 2.** We show that an isotropic initialization of $\mathbf{W}^{(k)}$ remains isotropic under gradient descent. The preservation of isotropy enforces alignment between $\mathbf{p}_j$ and $\mathbf{q}_j$ in expectation, i.e., $\mathbb{E}[\mathbf{p}_j^{(k,t)}] = \mathbb{E}[\mathbf{q}_j^{(k,t)}]$. Therefore, the posterior-difference term vanishes. **Step 3.** We analyze $\nabla \mathbf{p}_j^{(k,t)}$ and $\nabla \mathbf{q}_j^{(k,t)}$ based on the their inherent symmetric structure. This analysis shows the Jacobian difference term degenerates to the following symmetric matrix under expectation: $\left( \mathrm{diag}(1/d) - \frac{1}{d^2} \mathbf{1}\mathbf{1}^\top \right) \mathbf{M}^\top \left( \mathbf{W}^{(k)} - \mathbf{\Sigma}^{-1} \right)$, which enables us to avoid complicated analysis directly on the Jacobian difference term. Combining Steps 2 and 3, we obtain the following upper bound for the inner product term $-\langle \mathbf{W}^{(k)} - \mathbf{\Sigma}^{-1}, \nabla \mathcal{L} \rangle \leq -\alpha' \|\mathbf{W}^{(k)} - \mathbf{\Sigma}^{-1}\|_F^2$, which provides the desired result. The detailed proof can be found in Appendix D. $\square$

## 6 Experimental Results

**Compute resources.** All experiments are conducted on an NVIDIA H100 GPU with 80 GB of memory. The experiments require roughly five hours to complete.

**Problem setup.** In the following experiments, the augmented ICL instances are generated as follows. We set the number of classes $C = 3$ and the data dimension $d = 3$. The class mean vectors $\{\boldsymbol{\mu}_i\}_{i=1}^C$ are randomly sampled from a $d$-dimensional standard normal distribution. The covariance matrix $\Sigma = \epsilon \mathbf{I}_d$ is shared across classes, where $\mathbf{I}_d$ is the $d$-dimensional identity matrix. We set $\epsilon \in \{0.7, 1.5\}$. Each instance contains $N = 5$ labeled data points and $M$ unlabeled data points, where $M \in \{1, 10, 20\}$. The $M = 1$ case recovers the conventional ICL setting.

**Transformer structure.** We construct a transformer with the architecture specified in Theorem 4.1. This model features 4 layers, with each layer composed of an attention module followed by an MLP module. Activation functions for the attention layers are configured as follows: softmax for the first layer, linear for the second and third layers, and ReLU for the fourth layer. We set $d_p = 16$, and the number of CoT steps $T = 5$. During training, in each iteration, we randomly generate $64$ augmented ICL instances, and perform one gradient descent (GD) on the average empirical CoT training loss defined in Equation (5.1) over the batch. In total, we perform $15,000$ GD iterations during training.

**Results.** We evaluate the performance of the trained transformer after every 100 GD iterations. For evaluation, we randomly generated 100 augmented ICL instances, and obtained the corresponding class mean estimates from the trained transformer through CoT prompting. We then utilize these estimated class means to obtain the label prediction results according to Equation (3.5). For each $M \in \{1, 10, 20\}$, we conduct 5 runs. We track the the class mean estimation error and prediction accuracy, and plot the average performance and standard deviation across these 5 runs in Figure 1.

From Figure 1, we observe that augmented ICL outperforms conventional ICL significantly after a sufficient number of training iterations. As $M$ increases, the advantage becomes more prominent: the transformer's class mean estimation error decrease and the classification accuracy increase, as predicted by our theoretical results Theorem 4.2 and Corollary 4.1.

We notice that the advantage of augmented ICL is more significant when $\epsilon$ is relatively small. This is because when $\epsilon$ is small, the data distribution is less noisy, meaning that the features carry more information relevant to the labels. Therefore, the unlabeled data provides clearer structure that the transformer can leverage through augmented ICL to estimate class means more accurately.

## 7 Conclusion

In this work, we introduced augmented ICL, a framework in which models process a mixture of labeled and unlabeled examples within the prompt. We provided theoretical insights showing that transformers equipped with CoT reasoning can implement an EM-style algorithm for augmented ICL in a multi-class linear classification task, with provably decreasing prediction error as the amount of unlabeled data increases. Moreover, we showed that such transformer behavior can emerge through standard teacher forcing training. Our empirical results support the theory.

## Acknowledgments

The authors thank Li Fan, Wei Shen, Hadi Daneshmand and Cong Shen for their helpful discussions during the preparation and finalization of this work. RL and JY were partially supported by the U.S. NSF under grants 2318759, 2531023 and 2531789.

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

# Supplementary Materials

## Contents

## A  Broader Impacts

This work provides theoretical insights into how transformers can leverage unlabeled data to improve in-context learning, a core capability underlying many recent advances in language models. By improving data efficiency and adaptability, our findings could enable more accessible and capable AI systems, particularly in low-resource settings where labeled data is limited. These advances may benefit a range of applications, including next-generation wireless communications and networking, healthcare, and financial services. Given the theoretical nature of this work, we anticipate minimal direct negative societal impact. Nonetheless, we recognize that future practical implementations inspired by this research should adhere to responsible AI principles.

## B  Limitations and Future Directions

Our analysis and experiments possess certain limitations. Below, we outline these limitations and propose directions for future research.

First, our analysis tracks parameter updates only in the *first* transformer layer, leaving all other layers frozen. As a result, it remains unclear how weights in non-linear hidden layers evolve under teacher forcing. To the best of our knowledge, the training dynamics of multi-layer transformer with *non-linear* activation is still lacking investigation. A full, multi-layer treatment for end-to-end training remains an open problem.

Second, this paper is the first theoretical investigation of the influence of unlabeled data in in-context learning, therefore, we restricted the experiments to a synthetic data set. However, whether the same behavior emerges in real-world tasks, and how unlabeled examples influence in-context learning for large, fully-trained transformers, is still unknown. Empirically understanding such impact is a promising future direction.

## C  Proof of Expressiveness

### C.1  Proof of Theorem 4.1

We start from the proof of Theorem 4.1, which shows the transformer's capability of implementing an EM-style algorithm.

First, we restate the theorem below.

**Theorem C.1.** *There exists a 4-layer transformer, such that its output sequence at the $(t+1)$-th CoT step satisfies*

$$\widehat{\boldsymbol{\mu}}_i^{(t+1)} = \widehat{\boldsymbol{\mu}}_i^{(t)} - \frac{\eta^{(t)}}{M} \sum_{j=N+1}^{N+M} p_{ij}^{(t)} \left( \widehat{\boldsymbol{\mu}}_i^{(t)} - \mathbf{x}_j \right) + \mathbf{1}_{\{t=0\}} \cdot \frac{C}{N} \sum_{j=1}^{N} (\mathbf{e}_i^\top \mathbf{y}_j) \mathbf{x}_j, \tag{C.1}$$

*for any $i \in [C]$, where $\eta^{(t)} = \alpha/(T'+t)$ for some positive constants $\alpha$ and $T'$, $p_{ij}^{(t)}$ is the normalized weight*

$$p_{ij}^{(t)} = \frac{\sum_{\tau=0}^{t} \exp\left(-\frac{1}{2}\|\widehat{\boldsymbol{\mu}}_i^{(\tau)} - \mathbf{x}_j\|_{\boldsymbol{\Sigma}^{-1}}^2 + \beta\tau\right)}{\sum_{\tau=0}^{t} \sum_{c=1}^{C} \exp\left(-\frac{1}{2}\|\widehat{\boldsymbol{\mu}}_c^{(\tau)} - \mathbf{x}_j\|_{\boldsymbol{\Sigma}^{-1}}^2 + \beta\tau\right)}, \tag{C.2}$$

*and $\beta$ is a positive constant.*

Recall that the input sequence at the $t$-th CoT step is formulated as

$$\widehat{\mathbf{H}}^{(t-1)} = \begin{bmatrix} \mathbf{X}_\ell & \mathbf{X}_u & \mathbf{0} & \star & \cdots & \star \\ \mathbf{Y}_\ell & \mathbf{0} & \mathbf{0} & \star & \cdots & \star \\ \mathbf{P}_\ell & \mathbf{P}_u & \mathbf{Q}^{(0)} & \mathbf{Q}^{(1)} & \cdots & \mathbf{Q}^{(t-1)} \end{bmatrix},$$

where

$$\mathbf{P}_\ell = [\mathbf{p}_1, \quad \mathbf{p}_2, \quad \cdots \quad \mathbf{p}_N], \tag{C.3}$$

$$\mathbf{P}_u = [\mathbf{p}_{N+1}, \quad \mathbf{p}_{N+2}, \quad \cdots \quad \mathbf{p}_{N+M}], \tag{C.4}$$

$$\mathbf{Q}^{(\tau)} = [\mathbf{q}_1^{(\tau)}, \quad \mathbf{q}_2^{(\tau)}, \quad \cdots \quad \mathbf{q}_C^{(\tau)}], \quad \tau \in [0:t-1]. \tag{C.5}$$

We specify $\mathbf{p}_j$ and $\mathbf{q}_i^{(\tau)}$ as follows. For each data sample $j \in [N + M]$, we denote

$$
\mathbf{p}_j = \begin{bmatrix} \mathbf{0}_C \\ \mathbf{0}_d \\ \mathbf{0}_C \\ \mathbf{0}_C \\ 0 \\ \mathbf{1}_{j \in [N]} \\ \mathbf{1}_{j \in [N+1:N+M]} \\ 0 \end{bmatrix}, \quad \mathbf{q}_i^{(\tau)} = \begin{bmatrix} \mathbf{e}_i \\ \widehat{\boldsymbol{\mu}}_i^{(\tau)} \\ \mathbf{0}_C \\ \mathbf{0}_C \\ u_i^{(\tau)} \\ 0 \\ 0 \\ \tau \end{bmatrix},
$$

where $\widehat{\boldsymbol{\mu}}_i^{(\tau)}$ stores the estimate of the mean vector of class $i$ from the $\tau$-th CoT step, and $u_i^\tau$ stores a rescaled norm of $\widehat{\boldsymbol{\mu}}_i^{(\tau)}$, i.e., $u_i^{(\tau)} = -\frac{\sigma}{2}\|\widehat{\boldsymbol{\mu}}_i^{(\tau)}\|^2$.

Next, we specify the parameters of each layer of the transformer as follows.

**Layer 1:** The first layer of the transformer consists of an attention layer with a softmax activation function, and an MLP layer. Let the parameters of the attention layer satisfy

$$
\mathbf{Q}_1 \mathbf{K}_1 = \begin{bmatrix} \mathbf{0}_{d \times (d+2C)} & \boldsymbol{\Sigma}^{-1} & & & \\ & & \mathbf{0}_{(4C+d+2) \times 2C} & & \\ & & & 1 & \mathbf{0}_{1 \times 2} & \beta \\ & & & & & 0 \end{bmatrix},
$$

$$
\mathbf{V}_1 = \begin{bmatrix} \mathbf{0}_{(d+2C) \times (d+C)} & & \\ & \mathbf{I}_C & \\ & & \mathbf{0}_{(d+C+4) \times (d+2C+4)} \end{bmatrix}.
$$

Denote $\mathrm{attn}_1(\mathbf{p}_j)$ as the output token after passing $\mathbf{p}_j$ through the first attention layer, and let $\boldsymbol{\gamma}_i := \mathrm{attn}_1(\mathbf{p}_j)[d + C + 1 : d + 2C]$. Then, we have

$$
\boldsymbol{\gamma}_j = \frac{\sum_{\tau \in [0:t-1]} \sum_{i \in [C]} \exp\left(-\frac{\sigma}{2}\|\widehat{\boldsymbol{\mu}}_i^{(\tau)}\|_{\boldsymbol{\Sigma}^{-1}}^2 + (\widehat{\boldsymbol{\mu}}_i^{(\tau)})^\top \mathbf{x}_j + \beta\tau\right) \mathbf{e}_i}{\sum_{\tau \in [0:t-1]} \sum_{i \in [C]} \exp\left(-\frac{\sigma}{2}\|\widehat{\boldsymbol{\mu}}_i^{(\tau)}\|_{\boldsymbol{\Sigma}^{-1}}^2 + (\widehat{\boldsymbol{\mu}}_i^{(\tau)})^\top \mathbf{x}_j + \beta\tau\right)}.
$$

Other entries in $\widehat{\mathbf{H}}^{(t-1)}$ remain unchanged after this attention layer.

Subsequent to the first attention layer, a token-wise MLP is applied. Similar to Kim and Suzuki (2025), in this work, we assume the MLP layer can realize any deterministic token-wise link function with negligible error. The first MLP layer transforms input representations $\mathbf{p}$ such that

$$
\mathrm{mlp}_1(\mathrm{attn}_1(\mathbf{p}_j)) = \boldsymbol{\gamma}_j \cdot \mathrm{attn}_1(\mathbf{p}_j)[3C + d + 3]
$$
$$
\mathrm{mlp}_1(u_i^{(\tau)}) = -\frac{\sigma}{2}\|\widehat{\boldsymbol{\mu}}_i^{(\tau)}\|^2.
$$

Since $\mathbf{p}_j[3C + d + 3] = 0$ for $j \in [N]$ and $\mathbf{p}_j[3C + d + 3] = 1$ for $j \in [N + 1 : N + M]$, and the corresponding entries remain unchanged after passing through the first attention layer, this MLP layer only keeps $\boldsymbol{\gamma}_j$ for tokens corresponding to the unlabeled dataset (i.e., $j \in [N]$), and sets $\boldsymbol{\gamma}_j$ to zero for all other tokens (i.e., $j \in [N + 1 : M]$).

**Layer 2:** The second layer of the transformer consists of an attention layer with a linear activation function, and an MLP layer. The parameters of the attention layer are set to satisfy

$$
\mathbf{Q}_2 \mathbf{K}_2 = \begin{bmatrix} \mathbf{0}_{(2d+4C+2) \times (2d+4C+2)} & & \\ & 0 & 0 \\ & \alpha_1 & 0 \end{bmatrix},
$$

$$
\mathbf{V}_2 = \begin{bmatrix} \mathbf{0}_{(2d+3C) \times (d+2C)} & & \\ & \mathbf{I}_C & \\ & & \mathbf{0}_{4 \times (d+C+4)} \end{bmatrix}.
$$

We denote $\mathbf{s}_i^{(\tau)} := \mathrm{attn}_2(\mathbf{q}_i^{(\tau)})[d + 2C + 1 : d + 3C]$ as the vector extracted from the output token after passing $\mathbf{q}_i^{(\tau)}$ through the second attention layer. Then, $\mathbf{s}_i^{(\tau)} = \tau \alpha_1 \sum_{j=N+1}^{N+M} \boldsymbol{\gamma}_j$, where $\alpha_1$ is a fixed scalar embedded in $\mathbf{Q}_2 \mathbf{K}_2$.

We let the subsequent MLP layer realize the following token-wise Lipschitz function:

$$\mathrm{mlp}_2(\widehat{\boldsymbol{\mu}}_i^{(\tau)}) = \widehat{\boldsymbol{\mu}}_i^{(\tau)} - \frac{1}{\tau(\tau+\alpha_2)}\widehat{\boldsymbol{\mu}}_i^{(\tau)}\mathbf{e}_i^\top \mathbf{s}_i^{(\tau)} = \widehat{\boldsymbol{\mu}}_i^{(\tau)} - \frac{\alpha_1}{\tau+\alpha_2}\widehat{\boldsymbol{\mu}}_i^{(\tau)}\mathbf{e}_i^\top \sum_{j\in[N+1]}^{N+M}\boldsymbol{\gamma}_j,$$

$$\mathrm{mlp}_2(\mathbf{e}_i) = \frac{\alpha_1}{\tau+\alpha_2}\mathbf{e}_i.$$

**Layer 3:** Similar to the second transformer layer, the third layer also consists of a linear attention layer and an MLP layer. Consider the following parameterization for the attention layer:

$$\mathbf{Q}_3\mathbf{K}_3 = \begin{bmatrix} \mathbf{0}_{(d+C)\times(d+2C)} & & \\ & \mathbf{I}_C & \\ & & \mathbf{0}_{(d+2C+4)\times(d+C+4)} \end{bmatrix},$$

$$\mathbf{V}_3 = \begin{bmatrix} \mathbf{0}_{(d+3C)\times d} & \\ \mathbf{I}_d & \\ & \mathbf{0}_{C+4;d+4C+4} \end{bmatrix}.$$

Therefore, this attention layer realizes the following updating process:

$$\mathrm{attn}_3(\widehat{\boldsymbol{\mu}}_i^{(\tau)}) = \mathrm{mlp}_2(\widehat{\boldsymbol{\mu}}_i^{(\tau)}) + \frac{\alpha_1}{\tau+\alpha_2}\sum_{j\in[N+1,N+M]}\mathbf{x}_j\mathbf{e}_i^\top\boldsymbol{\gamma}_j^{(\tau)}.$$

After this linear attention layer, we let the MLP layer realize the following function

$$\mathrm{mlp}_3(\mathbf{e}_i) = \frac{\tau+\alpha_2}{\alpha_1}\mathbf{e}_i.$$

**Layer 4:** For the last layer, we introduce a transformer layer with a ReLU-activated attention layer followed by an MLP layer. We parameterize the attention layer as:

$$\mathbf{Q}_4\mathbf{K}_4 = \begin{bmatrix} \mathbf{0}_{(d+C)\times d} & & & \\ & \mathbf{I}_C & & \\ & & \mathbf{0}_{(d+2C+1)\times(d+3C+3)} & \\ & & & 1 \\ & & & \mathbf{0}_2 \end{bmatrix},$$

$$\mathbf{V}_4 = \begin{bmatrix} \mathbf{0}_{(d+2C)\times d} & \\ \mathbf{I}_d & \\ & \mathbf{0}_{(2C+4)\times(4C+d+4)} \end{bmatrix}.$$

The corresponding updating rule of this layer gives

$$\mathrm{attn}_4(\boldsymbol{\mu}_i^{(\tau)}) = \mathrm{attn}_3(\boldsymbol{\mu}_i^{(\tau)}) + \frac{C}{N}\sum_{j\in[N]}\mathbf{x}_j\mathrm{ReLU}(-\tau+\mathbf{e}_i^\top\mathbf{y}_j).$$

Therefore, we can further reformulate it as

$$\mathrm{attn}_4(\boldsymbol{\mu}_i^{(\tau)}) = \begin{cases} \dfrac{C}{N}\sum_{j\in[N]}\mathbf{x}_j\cdot(\mathbf{e}_i^\top\mathbf{y}_j), & \text{if } \tau=0, \\ \mathrm{attn}_3(\boldsymbol{\mu}_\tau^{(\tau)}), & \text{if } \tau>0. \end{cases}$$

Given the above 4-layer transformer structure, by setting $\alpha_1 = \alpha/M$ and $\alpha_2 = T'$ for fixed $\alpha > 0$, $T' > 0$, the output sequence corresponding to the $\mathbf{Q}^{(t-1)}$ block in the input sequence that satisfies:

$$\widehat{\boldsymbol{\mu}}_i^{(t+1)} = \widehat{\boldsymbol{\mu}}_i^{(t)} - \frac{\eta^{(t)}}{M}\sum_{j=N+1}^{N+M}p_{ij}^{(t)}\left(\widehat{\boldsymbol{\mu}}_i^{(t)} - \mathbf{x}_j\right) + \mathbf{1}_{\{t=0\}}\cdot\frac{C}{N}\sum_{j=1}^{N}(\mathbf{e}_i^\top\mathbf{y}_j)\mathbf{x}_j, \qquad \text{(C.6)}$$

for any $i \in [C]$, where $\eta^{(t)} = \alpha/(T'+t)$ for some positive constants $\alpha$ and $T'$, $p_{ij}^{(t)}$ is the normalized weight

$$p_{ij}^{(t)} = \frac{\sum_{\tau=0}^{t}\exp\left(-\frac{1}{2}\|\widehat{\boldsymbol{\mu}}_i^{(\tau)} - \mathbf{x}_j\|_{\boldsymbol{\Sigma}^{-1}}^2 + \beta\tau\right)}{\sum_{\tau=0}^{t}\sum_{c=1}^{C}\exp\left(-\frac{1}{2}\|\widehat{\boldsymbol{\mu}}_c^{(\tau)} - \mathbf{x}_j\|_{\boldsymbol{\Sigma}^{-1}}^2 + \beta\tau\right)}.$$

The proof is thus complete.

## C.2  Proof of Theorem 4.2

In this section, we show the detailed proof of Theorem 4.2. We start by restating the theorem.

**Theorem C.2** (Class Mean Estimation Error). *Given the transformer described in Theorem 4.1, when $N \geq \frac{36\alpha^2 L^2}{c_1} \log 1/\epsilon$ and $M \geq \max\{36\alpha^2 L^2 K, \log^2(1/\epsilon)\}$, and $t \geq \max\{\sqrt[4]{M}, T'\}$, with probability at least $1 - \epsilon$, the output of the transformer after $t$ CoT steps satisfies*

$$\|\widehat{\mathbf{M}}^{(t)} - \mathbf{M}\|_F^2 \leq c \frac{\log(1/\epsilon)}{N + \sqrt[4]{M}},$$

*where $c_1, c, \alpha, L, T', K$ are positive constants.*

**Step 1: First, we ensure that the initial estimation of the class mean vectors obtained from the** *labeled* **data gives a small estimation error.**

**Lemma 1** (Initial estimation error from labeled data). *Consider the initial class mean estimates*

$$\boldsymbol{\mu}_i^{(1)} = \frac{C}{N} \sum_{j \in [N]} \mathbf{x}_j \cdot (\mathbf{e}_i^\top \mathbf{y}_j), \quad \forall i \in [C].$$

*Then, for fixed $K \geq 1$ and any positive constant $T' \geq 4K$, we have*

$$\mathbb{P}\left[\left\|\boldsymbol{\mu}_i^{(1)} - \boldsymbol{\mu}_i\right\|^2 > \frac{K}{T'}\right] \leq \exp(-cNK/T'),$$

*where $c$ is a positive constant.*

*Proof.* We denote $n_i$ as the number of samples drawn from class $i$ in the $N$ labeled data. Under the assumption that $\mathbf{y}_j \sim \text{Uniform}(\mathcal{Y})$, $\forall j \in [N]$, we have $n_i \sim \text{Binomial}(N, 1/C)$. Then, according to the Chernoff's inequality, for any $\epsilon \in (0, 1)$, we have

$$\mathbb{P}\left(\left|n_i - \tfrac{N}{C}\right| > \epsilon \tfrac{N}{C}\right) \leq 2 \exp\left(-t \frac{\epsilon^2 N}{3C}\right).$$

For any $u \geq 0$, let $\epsilon = u\sqrt{K/T'}$, we obtain

$$\mathbb{P}\left(\left|n_i - \tfrac{N}{C}\right| > u\sqrt{\tfrac{K}{T'}} \tfrac{N}{C}\right) \leq 2 \exp\left(-\tfrac{u^2 NK}{3CT'}\right).$$

Therefore,

$$\mathbb{P}\left(\left|\tfrac{C}{N} n_i - 1\right| > u\sqrt{\tfrac{K}{T'}}\right) \leq 2 \exp\left(-\tfrac{u^2 NK}{3CT'}\right). \tag{C.7}$$

Conditional on $n_i$, we have $\frac{1}{n_i} \sum_{j:\mathbf{y}_i=\mathbf{e}_i} \mathbf{x}_j - \boldsymbol{\mu}_i \sim \mathcal{N}(0, \boldsymbol{\Sigma}/n_i)$. We assume $\boldsymbol{\Sigma}$ is an isotropic matrix in the form of $\sigma^2 \mathbb{1}$. Then, $\|\boldsymbol{\Sigma}\|_2 = \sigma^2$, and we obtain the following inequality based on the Hoeffding's inequality.

$$\mathbb{P}\left(\left\|\frac{1}{n_i} \sum_{j:\mathbf{y}_j=\mathbf{e}_i} \mathbf{x}_j - \boldsymbol{\mu}_i\right\| > \sigma\sqrt{\frac{2t}{n_i}} \Big| n_i\right) \leq 2e^{-t}.$$

For any $v \geq 0$, by setting $t = v^2 n_i K/(2\sigma^2 T')$, we have

$$\mathbb{P}\left(\left\|\frac{1}{n_i} \sum_{j:\mathbf{y}_j=\mathbf{e}_i} \mathbf{x}_j - \boldsymbol{\mu}_i\right\| > v\sqrt{\tfrac{K}{T'}}\right) \leq 2 \exp(-v^2 n_i K/(8\sigma^2 T'))$$

$$\leq 2 \exp\left(-v^2(1 - \tfrac{K}{T'}) \frac{NK}{C\sigma^2 T'}\right)$$

$$\leq 2 \exp\left(-v^2 \frac{NK}{2C\sigma^2 T'}\right). \tag{C.8}$$

Then,

$$
\mathbb{P}\Big(\|\widehat{\boldsymbol{\mu}}_i - \boldsymbol{\mu}_i\|^2 > \tfrac{K}{T'}\Big) = \mathbb{P}\Big(\Big\|\tfrac{C}{N}\sum_{j:\mathbf{y}_j=\mathbf{e}_i}\mathbf{x}_j - \tfrac{Cn_i}{N}\boldsymbol{\mu}_i - (1 - \tfrac{Cn_i}{N})\boldsymbol{\mu}_i\Big\| > \sqrt{\tfrac{K}{T'}}\Big)
$$

$$
\leq \mathbb{P}\Big(\tfrac{Cn_i}{N}\Big\|\tfrac{1}{n_i}\sum_{j:\mathbf{y}_j=\mathbf{e}_i}\mathbf{x}_j - \boldsymbol{\mu}_i\Big\| + |1 - \tfrac{Cn_i}{N}|\cdot\|\boldsymbol{\mu}_i\| \geq \sqrt{\tfrac{K}{T'}}\Big)
$$

$$
\leq \mathbb{P}\Big(\tfrac{Cn_i}{N}\Big\|\tfrac{1}{n_i}\sum_{j:\mathbf{y}_j=\mathbf{e}_i}\mathbf{x}_j - \boldsymbol{\mu}_i\Big\| \geq \sqrt{\tfrac{K}{T'}},\ \text{or}\ |1 - \tfrac{Cn_i}{N}|\cdot\|\boldsymbol{\mu}_i\| \geq \sqrt{\tfrac{K}{T'}}\Big)
$$

$$
\overset{(a)}{\leq} 4\exp\Big(-c\tfrac{NK}{T'}\Big)
$$

for positive constant $c$. The inequality $(a)$ holds by setting $u = 1/\|\boldsymbol{\mu}_i\|$ in Equation (C.8) and setting $v = N/Cn_1$ in Equation (C.22). The proof is thus complete. $\qquad\square$

**Step 2: Next, we bound the discrepancy between the gradient obtained from each CoT step for a given input sequence, and the gradient of the population loss.**

We define the population loss for any given set of class mean vectors $\{\boldsymbol{\mu}_i\}_{i\in[C]}$ (i.e., any given $\mathbf{M}$) as:

$$
\mathcal{L}(\{\boldsymbol{\mu}_i\}) = \mathbb{E}_{\mathbf{x}}\Big[\log\Big(\tfrac{1}{C}\sum_{i=1}^{C}\exp\big(-\tfrac{1}{2}\|\mathbf{x} - \boldsymbol{\mu}_i\|_{\boldsymbol{\Sigma}^{-1}}^2\big)\Big)\Big], \tag{C.9}
$$

where the expectation is taken over the randomly generated data $\mathbf{x}$ for given $\mathbf{M}$, as specified in Equation (3.1).

We first characterize an important property of $\mathcal{L}(\{\boldsymbol{\mu}_i\})$ as follows.

**Lemma 2.** *The Jacobian of $\nabla_{\boldsymbol{\mu}_i}\mathcal{L}$ at $\boldsymbol{\mu}_i$ for all $i \in [C]$ is negative definite, i.e., $\nabla_{\boldsymbol{\mu}_i}^2\mathcal{L} \prec \mathbf{0}$.*

*Proof.* Define

$$
p_{\mathbf{x}}(\boldsymbol{\mu}_i) = \frac{\exp\Big(-\tfrac{1}{2}\|\mathbf{x} - \boldsymbol{\mu}_i\|_{\boldsymbol{\Sigma}^{-1}}^2\Big)}{\sum_{c=1}^{C}\exp\Big(-\tfrac{1}{2}\|\mathbf{x} - \boldsymbol{\mu}_c\|_{\boldsymbol{\Sigma}^{-1}}^2\Big)},
$$

so that $p_{\mathbf{x}}(\boldsymbol{\mu}_i)$ is a softmax weight depending on $\mathbf{x}$ and the centers $\{\boldsymbol{\mu}_c\}_{c=1}^{C}$. Note that $\nabla_{\boldsymbol{\mu}_i}^2\mathcal{L}$ is the Hessian of $\nabla\mathcal{L}$ at $\boldsymbol{\mu}_i$, given by

$$
\nabla_{\boldsymbol{\mu}_i}^2\mathcal{L} = \mathbb{E}_{\mathbf{x}}\Big[p_{\mathbf{x}}(\boldsymbol{\mu}_i)\big(1 - p_{\mathbf{x}}(\boldsymbol{\mu}_i)\big)\boldsymbol{\Sigma}^{-1}(\boldsymbol{\mu}_i - \mathbf{x})(\boldsymbol{\mu}_i - \mathbf{x})^{\top}\boldsymbol{\Sigma}^{-1} - p_{\mathbf{x}}(\boldsymbol{\mu}_i)\boldsymbol{\Sigma}^{-1}\Big],
$$

where and the expectation is taken with respect to the distribution of $\mathbf{x}$. Therefore, there exists a constant $0 \leq \alpha < 1$ such that

$$
\nabla_{\boldsymbol{\mu}_i}^2\mathcal{L} \preceq \mathbb{E}_{\mathbf{x}}\Big[\alpha\, p_{\mathbf{x}}(\boldsymbol{\mu}_i)\boldsymbol{\Sigma}^{-1}(\boldsymbol{\mu}_i - \mathbf{x})(\boldsymbol{\mu}_i - \mathbf{x})^{\top}\boldsymbol{\Sigma}^{-1} - p_{\mathbf{x}}(\boldsymbol{\mu}_i)\boldsymbol{\Sigma}^{-1}\Big].
$$

Now, for any nonzero vector $\mathbf{u} \in \mathbb{R}^d$, consider the quadratic form $\mathbf{u}^{\top}\nabla_{\boldsymbol{\mu}_i}^2\mathcal{L}\mathbf{u}$, using the above matrix inequality, we have

$$
\mathbf{u}^{\top}\nabla_{\boldsymbol{\mu}_i}^2\mathcal{L}\mathbf{u} \leq \mathbf{u}^{\top}\mathbb{E}_{\mathbf{x}}\Big[\alpha\, p_{\mathbf{x}}(\boldsymbol{\mu}_i)\boldsymbol{\Sigma}^{-1}(\boldsymbol{\mu}_i - \mathbf{x})(\boldsymbol{\mu}_i - \mathbf{x})^{\top}\boldsymbol{\Sigma}^{-1} - p_{\mathbf{x}}(\boldsymbol{\mu}_i)\boldsymbol{\Sigma}^{-1}\Big]\mathbf{u}.
$$

Therefore, rewriting the expectation as an integral yields

$$
\mathbf{u}^{\top}\nabla_{\boldsymbol{\mu}_i}^2\mathcal{L}\mathbf{u} \leq \frac{1}{C}\int_{\mathbb{R}^d}\alpha\,\mathcal{N}(\mathbf{x} \mid \boldsymbol{\mu}_i, \mathbf{I})\mathbf{u}^{\top}\boldsymbol{\Sigma}^{-1}(\boldsymbol{\mu}_i - \mathbf{x})(\boldsymbol{\mu}_i - \mathbf{x})^{\top}\boldsymbol{\Sigma}^{-1}\mathbf{u}\,\mathrm{d}\mathbf{x} - \frac{1}{C}\mathbf{u}^{\top}\boldsymbol{\Sigma}^{-1}\mathbf{u}
$$

$$
\leq \frac{\alpha}{C}\mathbf{u}^{\top}\boldsymbol{\Sigma}^{-1}\mathbb{E}_{\mathbf{x}\sim\mathcal{N}(\boldsymbol{\mu}_i, \boldsymbol{\Sigma})}\Big[(\boldsymbol{\mu}_i - \mathbf{x})(\boldsymbol{\mu}_i - \mathbf{x})^{\top}\Big]\boldsymbol{\Sigma}^{-1}\mathbf{u} - \frac{1}{C}\mathbf{u}^{\top}\boldsymbol{\Sigma}^{-1}\mathbf{u} < 0.
$$

Thus, the quadratic form is negative for every nonzero $\mathbf{u}$, and the matrix $\nabla_{\boldsymbol{\mu}_i}^2\mathcal{L}$ is negative definite. This completes the proof. $\qquad\square$

We note that for each CoT step $t > 0$, the updating induced by the constructed transformer is

$$\widehat{\boldsymbol{\mu}}_i^{(t+1)} = \widehat{\boldsymbol{\mu}}_i^{(t)} - \frac{\eta^{(t)}}{M} \sum_{j=N+1}^{N+M} p_{ij}^{(t)} \big(\widehat{\boldsymbol{\mu}}_i^{(t)} - \mathbf{x}_j\big), \tag{C.10}$$

where $p_{ij}^{(t)}$ is defined in Equation (4.2).

To simplify notation, denote

$$\frac{1}{M} \sum_{j=N+1}^{N+M} p_{ij}^{(t)} \big(\widehat{\boldsymbol{\mu}}_i^{(t)} - \mathbf{x}_j\big) := \nabla_{\widehat{\boldsymbol{\mu}}_i^{(t)}} \widehat{\mathcal{L}}. \tag{C.11}$$

We note that $\widehat{\mathcal{L}}$ itself is not an explicit loss function. We use the notation $\nabla_{\widehat{\boldsymbol{\mu}}_i^{(t)}} \widehat{\mathcal{L}}$ to represent the equivalent gradient for the updating determined by the $t$-th CoT step. Lemma 2 states that $\nabla_{\boldsymbol{\mu}_i}^2 \mathcal{L}$ is negative definite for each $\boldsymbol{\mu}_i$. In the following lemma, we show that $\nabla_{\boldsymbol{\mu}}^2 \mathcal{L}$ is negative definite for the concatenate vector $\boldsymbol{\mu}$ when $\{\boldsymbol{\mu}_i\}_i^C$ are well seperated.

**Lemma 3.** *The Jacobian of $\nabla_{\boldsymbol{\mu}} \mathcal{L}$ at $\boldsymbol{\mu}$ is negative definite, i.e., $\nabla_{\boldsymbol{\mu}}^2 \mathcal{L} \prec 0$.*

*Proof.* Recall that $\boldsymbol{\Sigma} \in \mathbb{R}^{d \times d}$ is a symmetric positive definite. For $\mathbf{x} \in \mathbb{R}^d$ and $\boldsymbol{\mu} = (\boldsymbol{\mu}_1, \dots, \boldsymbol{\mu}_C) \in (\mathbb{R}^d)^C$, define

$$\ell(\mathbf{x}; \boldsymbol{\mu}) = \log\Big(\tfrac{1}{C} \sum_{i=1}^{C} \exp\big(-\tfrac{1}{2}\|\mathbf{x} - \boldsymbol{\mu}_i\|_{\boldsymbol{\Sigma}^{-1}}^2\big)\Big).$$

Therefore, we have $\mathcal{L}(\boldsymbol{\mu}) = \mathbb{E}_{\mathbf{x}}\big[\ell(\mathbf{x}; \boldsymbol{\mu})\big]$. The quadratic form of the Hessian of $\ell$ in direction $\boldsymbol{\Delta} = (\boldsymbol{\Delta}_1, \dots, \boldsymbol{\Delta}_C)$ can be written as

$$\boldsymbol{\Delta}^\top \nabla^2 \ell(\mathbf{x}; \boldsymbol{\mu}) \, \boldsymbol{\Delta} = -\sum_{i=1}^{C} p_{\mathbf{x}}(\boldsymbol{\mu}_i) \|\boldsymbol{\Delta}_i\|_{\boldsymbol{\Sigma}^{-1}}^2 + \mathrm{Var}_{p_{\mathbf{x}}}\Big(\{\langle \boldsymbol{\Sigma}^{-1}(\mathbf{x} - \boldsymbol{\mu}_i), \boldsymbol{\Delta}_i\rangle\}_{i=1}^C\Big), \tag{C.12}$$

Define

$$\rho_{ij}^2 := \|\boldsymbol{\mu}_i - \boldsymbol{\mu}_j\|_{\boldsymbol{\Sigma}^{-1}}^2, \quad \rho_\star := \min_{i \neq j} \rho_{ij}, \quad \Delta_{ij}(\mathbf{x}) := \|\mathbf{x} - \boldsymbol{\mu}_j^\star\|_{\boldsymbol{\Sigma}^{-1}}^2 - \|\mathbf{x} - \boldsymbol{\mu}_i^\star\|_{\boldsymbol{\Sigma}^{-1}}^2.$$

Then, we obtain that

$$\Delta_{ij}(\mathbf{x}) = \rho_{ij}^2 + 2z_{ij},$$

where $z_{ij} \sim \mathcal{N}(0, \rho_{ij}^2)$.

Define the event

$$\mathcal{E} := \Big\{\min_{j \neq i} \Delta_{ij}(\mathbf{x}) \geq \tfrac{1}{2}\rho_\star^2\Big\}. \tag{C.13}$$

and its complement as $\bar{\mathcal{E}}$. Then, based on the Gaussian tail bound and taking a union bound over $j \neq i$, we have

$$\Pr(\mathcal{E}) \geq 1 - (C-1)e^{-\rho_\star^2/8} =: 1 - \eta_\star. \tag{C.14}$$

Then, under event $\mathcal{E}$, $p_{\mathbf{x}}(\boldsymbol{\mu}_i)$ is lower bounded by

$$p_{\mathbf{x}}(\boldsymbol{\mu}_i) \geq \frac{1}{1 + \sum_{j \neq i} e^{-\frac{1}{2}\Delta_{ij}(\mathbf{x})}} \geq \frac{1}{1 + (C-1)e^{-\rho_\star^2/4}} =: \beta_\star. \tag{C.15}$$

Using (C.14)–(C.15), we obtain

$$\mathbb{E}_{\mathbf{x}}\big[p_{\mathbf{x}}(\boldsymbol{\mu}_i)\big] = \mathbb{E}_{\mathbf{x}}\big[p_{\mathbf{x}}(\boldsymbol{\mu}_i)|\mathcal{E}\big]\Pr(\mathcal{E}) + \mathbb{E}_{\mathbf{x}}\big[p_{\mathbf{x}}(\boldsymbol{\mu}_i)|\bar{\mathcal{E}}\big] \geq (1 - \eta_\star)\beta_\star > 0. \tag{C.16}$$

Therefore, based on Equation (C.16), the expectation of the first term in Equation (C.12) can be upper bounded as

$$\mathbb{E}\Big[ -\sum_{i=1}^{C} p_{\mathbf{x}}(\boldsymbol{\mu}_i)\,\|\boldsymbol{\Delta}_i\|_{\boldsymbol{\Sigma}^{-1}}^2 \Big] \le -\sum_i (1-\eta_\star)\beta_\star \|\boldsymbol{\Delta}_i\|_{\boldsymbol{\Sigma}^{-1}}. \tag{C.17}$$

Define

$$u_i =: \langle \boldsymbol{\Sigma}^{-1}(\mathbf{x}-\boldsymbol{\mu}_i),\, \boldsymbol{\Delta}_i \rangle. \tag{C.18}$$

Then, the second term in Equation (C.12) can be expressed as

$$\mathrm{Var}_{p_{\mathbf{x}}}(\{u_i\}_{i=1}^C) = \sum_{i=1}^{C} p_{\mathbf{x}}(\boldsymbol{\mu}_i)u_i^2 - \Big(\sum_{i=1}^{C} p_{\mathbf{x}}(\boldsymbol{\mu}_i)u_i\Big)^2 \ge 0.$$

Note that

$$\begin{aligned}
\mathrm{Var}_{p_{\mathbf{x}}}(\{u_i\}_{i=1}^C) &= \sum_{i=1}^{C} p_{\mathbf{x}}(\boldsymbol{\mu}_i)u_i^2 - \Big(\sum_{i=1}^{C} p_{\mathbf{x}}(\boldsymbol{\mu}_i)u_i\Big)^2 \\
&= \Big(\sum_{i=1}^{C} p_{\mathbf{x}}(\boldsymbol{\mu}_i)u_i^2\Big)\Big(\sum_{i=1}^{C} p_{\mathbf{x}}(\boldsymbol{\mu}_i)\Big) - \Big(\sum_{i=1}^{C} p_{\mathbf{x}}(\boldsymbol{\mu}_i)u_i\Big)^2 \\
&= \frac{1}{2}\Big(\sum_{i,j} p_{\mathbf{x}}(\boldsymbol{\mu}_i)p_{\mathbf{x}}(\boldsymbol{\mu}_j)(u_i^2+u_j^2)\Big) - \Big(\sum_{i=1}^{C} p_{\mathbf{x}}(\boldsymbol{\mu}_i)u_i\Big)^2 \\
&= \sum_{i<j} p_{\mathbf{x}}(\boldsymbol{\mu}_i)p_{\mathbf{x}}(\boldsymbol{\mu}_j)(u_i-u_j)^2 \\
&\le \sum_{i<j} p_{\mathbf{x}}(\boldsymbol{\mu}_i)p_{\mathbf{x}}(\boldsymbol{\mu}_j)2(u_i^2+u_j^2) \\
&= 2\sum_{i}\sum_{j\ne i} p_{\mathbf{x}}(\boldsymbol{\mu}_i)p_{\mathbf{x}}(\boldsymbol{\mu}_j)u_i^2 \\
&= 2\sum_{i} p_{\mathbf{x}}(\boldsymbol{\mu}_i)(1-p_{\mathbf{x}}(\boldsymbol{\mu}_i))u_i^2. \tag{C.19}
\end{aligned}$$

Next, we condition $\mathrm{Var}_{p_{\mathbf{x}}}(\{u_i\}_{i=1}^C)$ on event $\mathcal{E}$ and its complement $\bar{\mathcal{E}}$, respectively. When event $\mathcal{E}$ holds, combining Equation (C.15) and Equation (C.19) gives

$$\begin{aligned}
\mathrm{Var}_{p_{\mathbf{x}}}(\{u_i\}_{i=1}^C|\mathcal{E}) &\le 2\sum_{i} p_{\mathbf{x}}(\boldsymbol{\mu}_i)(1-p_{\mathbf{x}}(\boldsymbol{\mu}_i))u_i^2 \\
&\le 2\sum_{i}(1-\beta_\star)u_i^2
\end{aligned}$$

Under event $\bar{\mathcal{E}}$, we have

$$\mathrm{Var}_{p_{\mathbf{x}}}(\{u_i\}_{i=1}^C|\bar{\mathcal{E}}) \le \frac{1}{2}\sum_i u_i^2.$$

Define

$$k_\star = \max_{1\le i\le C} \mathbb{E}_{\mathbf{x}}\big[\|\mathbf{x}-\boldsymbol{\mu}_i\|_{\boldsymbol{\Sigma}^{-1}}^2\big].$$

Then, based on the definition of $u_i$ in Equation (C.18), by using spectral norm inequality, we have

$$\mathbb{E}_{\mathbf{x}}\big[u_i^2\big] = \boldsymbol{\Delta}_i^\top \boldsymbol{\Sigma}^{-1}\, \mathbb{E}\big[(\mathbf{x}-\boldsymbol{\mu}_i)(\mathbf{x}-\boldsymbol{\mu}_i)^\top\big]\,\boldsymbol{\Sigma}^{-1}\,\boldsymbol{\Delta}_i \le k_\star \|\boldsymbol{\Delta}_i\|_{\boldsymbol{\Sigma}^{-1}}^2.$$

Taking expectation over $\mathbf{x}$, we obtain

$$\mathbb{E}_{\mathbf{x}}\big[\mathrm{Var}_{p_{\mathbf{x}}}(\{u_i\}_{i=1}^C)\big] \le \Big(2(1-\beta_\star)+\tfrac{\eta_\star}{2}\Big) k_\star \sum_i \|\boldsymbol{\Delta}_i\|_{\boldsymbol{\Sigma}^{-1}}^2, \tag{C.20}$$

Plugging Equation (C.17) and Equation (C.20) into the expectation of Equation (C.12), we have

$$\mathbb{E}_{\mathbf{x}}\left[\boldsymbol{\Delta}^{\top}\nabla^{2}\mathcal{L}(\boldsymbol{\mu})\,\boldsymbol{\Delta}\right] \leq -\left((1-\eta_{\star})\beta_{\star} - \left(2(1-\beta_{\star})+\tfrac{\eta_{\star}}{2}\right)k_{\star}\right)\sum_{i}\|\boldsymbol{\Delta}_{i}\|_{\boldsymbol{\Sigma}^{-1}}^{2}.$$

Recall that

$$\eta_{\star} = (C-1)e^{-\rho_{\star}^{2}/8}, \quad \beta_{\star} = \frac{1}{1+(C-1)e^{-\rho_{\star}^{2}/4}},$$

and $\rho_{\star}$ is defined as

$$\rho_{ij}^{2} := \|\boldsymbol{\mu}_{i}-\boldsymbol{\mu}_{j}\|_{\boldsymbol{\Sigma}^{-1}}^{2}, \quad \rho_{\star} := \min_{i\neq j}\rho_{ij}.$$

Then, for sufficiently large $\rho_{\star}$ such that

$$(1-\eta_{\star})\beta_{\star} - \left(2(1-\beta_{\star})+\tfrac{\eta_{\star}}{2}\right)k_{\star} > 0,$$

$\nabla^{2}\mathcal{L}(\boldsymbol{\mu})$ is negative definite .

$\square$

In the following, we characterize $\nabla_{\widehat{\boldsymbol{\mu}}_{i}^{(t)}}\widehat{\mathcal{L}}$ and compare it with $\nabla\mathcal{L}$, i.e., the gradient if GD is performed on the population loss. We have the following lemma.

**Lemma 4** (Properties of the CoT gradient descent). *Fix an epoch $t$ and a component index $i \in [C]$, there exist constants $c_{1}, c_{2} > 0$ such that, for every $M \geq 1$,*

$$\Pr\left(\left\|\nabla_{\widehat{\boldsymbol{\mu}}_{i}^{(t)}}\widehat{\mathcal{L}} - \nabla_{\widehat{\boldsymbol{\mu}}_{i}^{(t)}}\mathcal{L}\right\| \leq c_{1}\,M^{-1/4}\right) \geq 1 - \exp(-\sqrt{M}),$$

*and*

$$\Pr\left(\left\|\nabla_{\widehat{\boldsymbol{\mu}}_{i}^{(t)}}\widehat{\mathcal{L}}\right\|^{2} \leq c_{2} + c_{3}M^{-1/2}\right) \geq 1 - \exp(-\sqrt{M}).$$

*Proof.* Recall that

$$\nabla_{\widehat{\boldsymbol{\mu}}_{i}^{(t)}}\widehat{\mathcal{L}} = \frac{1}{M}\sum_{j=N+1}^{N+M}p_{ij}^{(t)}\big(\widehat{\boldsymbol{\mu}}_{i}^{(t)}-\mathbf{x}_{j}\big),$$

where $\widehat{p}_{ij}^{(k,t)}$ is given by

$$p_{ij}^{(t)} = \frac{\sum_{\tau=0}^{t}\exp\left(-\frac{1}{2}\|\widehat{\boldsymbol{\mu}}_{i}^{(\tau)}-\mathbf{x}_{j}\|_{\boldsymbol{\Sigma}^{-1}}^{2}+\beta\tau\right)}{\sum_{\tau=0}^{t}\sum_{c=1}^{C}\exp\left(-\frac{1}{2}\|\widehat{\boldsymbol{\mu}}_{c}^{(\tau)}-\mathbf{x}_{j}\|_{\boldsymbol{\Sigma}^{-1}}^{2}+\beta\tau\right)}.$$

By choosing $\beta \to \infty$, we further have

$$p_{ij}^{(t)} = \frac{\exp\left(-\frac{1}{2}\|\widehat{\boldsymbol{\mu}}_{i}^{(\tau)}-\mathbf{x}_{j}\|_{\boldsymbol{\Sigma}^{-1}}^{2}\right)}{\sum_{c=1}^{C}\exp\left(-\frac{1}{2}\|\widehat{\boldsymbol{\mu}}_{c}^{(\tau)}-\mathbf{x}_{j}\|_{\boldsymbol{\Sigma}^{-1}}^{2}\right)}.$$

where the samples $\{\mathbf{x}_{j}\}_{j\geq N+1}$ are drawn from a Gaussian mixture distribution.

Therefore, given $\widehat{\boldsymbol{\mu}}_{ij}^{(t)}$, the random variable $p_{ij}^{(t)}\big(\widehat{\boldsymbol{\mu}}_{i}^{(t)}-\mathbf{x}_{j}\big)$ admits a sub-Gaussian tail bound since $\mathbf{x}_{j}$ are Guassian random vectors and $p_{ij}^{(t)}\big(\widehat{\boldsymbol{\mu}}_{i}^{(t)}-\mathbf{x}_{j}\big)$ is Lipschitz continuous over $\mathbf{x}_{j}$.

Then, by the Bernstein's inequality, for any fixed $\delta \in (0,1)$, with probability at least $1-\delta$, we have

$$\left\|\nabla_{\widehat{\boldsymbol{\mu}}_{i}^{(t)}}\widehat{\mathcal{L}} - \nabla_{\widehat{\boldsymbol{\mu}}_{i}^{(t)}}\mathcal{L}\right\| = \left\|\frac{1}{M}\sum_{j=N+1}^{N+M}p_{ij}^{(t)}\big(\widehat{\boldsymbol{\mu}}_{i}^{(t)}-\mathbf{x}_{j}\big) - \mathbb{E}_{\mathbf{x}_{j}}\left[p_{ij}^{(t)}\big(\widehat{\boldsymbol{\mu}}_{i}^{(t)}-\mathbf{x}_{j}\big)\right]\right\|$$

$$\leq \frac{c_{4}}{\sqrt{M}}\sqrt{\log\left(\tfrac{2}{\delta}\right)},$$

where $c_4 > 0$ is some absolute constant.

By choosing $\delta = \exp(-\sqrt{M})$, we obtain that with probability at least $1 - \exp(-\sqrt{M})$,

$$\left\|\nabla_{\widehat{\boldsymbol{\mu}}_i^{(t)}}\widehat{\mathcal{L}} - \nabla_{\widehat{\boldsymbol{\mu}}_i^{(t)}}\mathcal{L}\right\| \le c' M^{-\frac{1}{4}}. \tag{C.21}$$

for another constant $c' > 0$. This completes the proof of the first inequality.

Next, we show that $\|\nabla_{\widehat{\boldsymbol{\mu}}_i^{(t)}}\widehat{\mathcal{L}}\|$ itself is bounded with high probability.

Consequently,

$$\begin{aligned}
\|\nabla_{\widehat{\boldsymbol{\mu}}_i^{(t)}}\widehat{\mathcal{L}}\| &= \left\|\frac{1}{M}\sum_{j=N+1}^{N+M} p_{ij}^{(t)}\left(\widehat{\boldsymbol{\mu}}_i^{(t)} - \mathbf{x}_j\right)\right\| \\
&\le \frac{1}{M}\sum_{j=N+1}^{N+M}\left\|p_{ij}^{(t)}\left(\widehat{\boldsymbol{\mu}}_i^{(t)} - \mathbf{x}_j\right)\right\| \\
&\overset{(a)}{\le} \frac{1}{M}\sum_{j\ge N+1}\left\|\widehat{\boldsymbol{\mu}}_i^{(t)} - \mathbf{x}_j\right\| \\
&\le \frac{1}{M}\sum_{j\ge N+1}\left(\|\widehat{\boldsymbol{\mu}}_i^{(t)}\| + \|\mathbf{x}_j\|\right) \\
&= \|\widehat{\boldsymbol{\mu}}_i^{(t)}\| + \frac{1}{M}\sum_{j\ge N+1}\|\mathbf{x}_j\|, \tag{C.22}
\end{aligned}$$

where inequality $(a)$ holds since $p_{ij}^{(t)} \le 1$. Note that

$$\begin{aligned}
\widehat{\boldsymbol{\mu}}_i^{(t)} &= \widehat{\boldsymbol{\mu}}_i^{(t-1)} - \frac{\eta^{(t-1)}}{M}\sum_{j=N+1}^{N+M} p_{ij}^{(t-1)}\left(\widehat{\boldsymbol{\mu}}_i^{(t-1)} - \mathbf{x}_j\right) \\
&= \left(1 - \frac{\eta^{(t-1)}}{M}\right)\widehat{\boldsymbol{\mu}}_i^{(t-1)} + \frac{\eta^{(t-1)}}{M}\sum_{j=N+1}^{N+M} p_{i,j}^{(t-1)}\mathbf{x}_j.
\end{aligned}$$

Therefore, we have

$$\begin{aligned}
\|\widehat{\boldsymbol{\mu}}_i^{(t)}\| &\le \|\widehat{\boldsymbol{\mu}}_i^{(t-1)}\| + \frac{1}{M}\sum_{j\ge N+1}\|\mathbf{x}_j\| \\
&\le \|\widehat{\boldsymbol{\mu}}_i^{(1)}\| + \frac{t-1}{M}\sum_{j\ge N+1}\|\mathbf{x}_j\|
\end{aligned}$$

Combining with Equation (C.22), we have

$$\left\|\nabla_{\widehat{\boldsymbol{\mu}}_i^{(t)}}\widehat{\mathcal{L}}\right\| \le \|\widehat{\boldsymbol{\mu}}_i^{(1)}\| + \frac{t}{M}\sum_{j\ge N+1}\|\mathbf{x}_j\|.$$

Applying the Bernstein's inequality, with probability at least $1 - \exp(-\sqrt{M})$, we have

$$\frac{1}{M}\sum_{j\ge N+1}\|\mathbf{x}_j\| \le \frac{1}{C}\sum_{i=1}^{C}\boldsymbol{\mu}_i + c_5 M^{-\frac{1}{4}},$$

where $c_5$ is a positive constant.

Therefore, for any $t \le T$ where $T$ is total number of CoT steps, we have

$$\left\|\nabla_{\widehat{\boldsymbol{\mu}}_i^{(t)}}\widehat{\mathcal{L}}\right\| \le \|\widehat{\boldsymbol{\mu}}_i^{(1)}\| + \frac{T}{C}\sum_{i=1}^{C}\boldsymbol{\mu}_i + c_5 t M^{-\frac{1}{4}},$$

which implies

$$\|\nabla_{\widehat{\boldsymbol{\mu}}_i^{(t)}}\widehat{\mathcal{L}}\|^2 \le c_2 + c_3 M^{-\frac{1}{2}},$$

where $c_2$ and $c_3$ are positive constants depends on $T$, $\|\widehat{\boldsymbol{\mu}}_i^{(1)}\|$ and $\frac{T}{C}\sum_{i=1}^{C}\boldsymbol{\mu}_i$. The proof is thus complete. $\qquad\square$

**Step 3: Finally, we show the convergence of the class mean estimation error.**

Expanding the squared error $\|\widehat{\boldsymbol{\mu}}_i^{(t+1)} - \boldsymbol{\mu}_i\|^2$ gives

$$\begin{aligned}
\|\widehat{\boldsymbol{\mu}}_i^{(t+1)} - \boldsymbol{\mu}_i\|^2 =& \|\widehat{\boldsymbol{\mu}}_i^{(t)} - \boldsymbol{\mu}_i\|^2 + 2\,\eta^{(t)}\Big\langle\widehat{\boldsymbol{\mu}}_i^{(t)} - \boldsymbol{\mu}_i,\, \nabla_{\widehat{\boldsymbol{\mu}}_i^{(t)}}\widehat{\mathcal{L}}\Big\rangle + (\eta^{(t)})^2\left\|\nabla_{\widehat{\boldsymbol{\mu}}_i^{(t)}}\widehat{\mathcal{L}}\right\|^2 \\
\le& \|\widehat{\boldsymbol{\mu}}_i^{(t)} - \boldsymbol{\mu}_i\|^2 + 2\,\eta^{(t)}\Big\langle\widehat{\boldsymbol{\mu}}_i^{(t)} - \boldsymbol{\mu}_i,\, \nabla_{\widehat{\boldsymbol{\mu}}_i^{(t)}}\mathcal{L}\Big\rangle + 2\,\eta^{(t)}\left\|\nabla\mathcal{L} - \nabla\widehat{\mathcal{L}}\right\| \\
& + (\eta^{(t)})^2\left\|\nabla_{\widehat{\boldsymbol{\mu}}_i^{(t)}}\widehat{\mathcal{L}}\right\|^2.
\end{aligned} \tag{C.23}$$

Denote $\widehat{\boldsymbol{\mu}}^{(t)}$ and $\boldsymbol{\mu}$ as the vectors obtained by stacking $\{\widehat{\boldsymbol{\mu}}_i^{(t)}\}_{i=1}^{C}$ and $\{\boldsymbol{\mu}_i\}_{i=1}^{C}$, respectively. Therefore, we have

$$\|\widehat{\boldsymbol{\mu}}^{(t+1)} - \boldsymbol{\mu}\|^2 \le \|\widehat{\boldsymbol{\mu}}^{(t)} - \boldsymbol{\mu}\|^2 + 2\,\eta^{(t)}\langle\widehat{\boldsymbol{\mu}}^{(t)} - \boldsymbol{\mu},\, \nabla_{\widehat{\boldsymbol{\mu}}^{(t)}}\mathcal{L}\rangle + 2\,\eta^{(t)}\left\|\nabla\mathcal{L} - \nabla\widehat{\mathcal{L}}\right\| + (\eta^{(t)})^2\left\|\nabla_{\widehat{\boldsymbol{\mu}}^{(t)}}\widehat{\mathcal{L}}\right\|^2.$$

To control the inner product term $\langle\widehat{\boldsymbol{\mu}}^{(t)} - \boldsymbol{\mu},\, \nabla_{\widehat{\boldsymbol{\mu}}^{(t)}}\mathcal{L}\rangle$, we perform a first-order Taylor expansion of $\nabla_{\widehat{\boldsymbol{\mu}}^{(t)}}\mathcal{L}$ around $\boldsymbol{\mu}$ as

$$\begin{aligned}
\nabla_{\widehat{\boldsymbol{\mu}}^{(t)}}\mathcal{L} &= \nabla_{\boldsymbol{\mu}}\mathcal{L} + (\nabla_{\boldsymbol{\mu}}^2\mathcal{L})(\widehat{\boldsymbol{\mu}}^{(t)} - \boldsymbol{\mu}) + \mathbf{R}(\widehat{\boldsymbol{\mu}}^{(t)}, \boldsymbol{\mu}) \\
&\overset{(a)}{=} (\nabla_{\boldsymbol{\mu}}^2\mathcal{L})(\widehat{\boldsymbol{\mu}}^{(t)} - \boldsymbol{\mu}) + \mathbf{R}(\widehat{\boldsymbol{\mu}}^{(t)}, \boldsymbol{\mu}),
\end{aligned}$$

where equality $(a)$ holds since $\boldsymbol{\mu}$ is the global minimizer of $\mathcal{L}$ and $\mathcal{L}$ is differentiable on $\mathbb{R}^d$, thus $\nabla_{\boldsymbol{\mu}}\mathcal{L} = 0$, and $\mathbf{R}(\widehat{\boldsymbol{\mu}}^{(t)}, \boldsymbol{\mu})$ is the remainder term.

For the remainder term, we have

$$\begin{aligned}
&\langle\mathbf{R}(\widehat{\boldsymbol{\mu}}^{(t)}, \boldsymbol{\mu}), \widehat{\boldsymbol{\mu}}^{(t)} - \boldsymbol{\mu}\rangle \\
&= \int_0^1 (\widehat{\boldsymbol{\mu}}^{(t)} - \boldsymbol{\mu})^\top\Big(\nabla_{\boldsymbol{\mu}+\xi(\widehat{\boldsymbol{\mu}}^{(t)}-\boldsymbol{\mu})}^2\mathcal{L} - \nabla_{\boldsymbol{\mu}}^2\mathcal{L}\Big)(\widehat{\boldsymbol{\mu}}^{(t)} - \boldsymbol{\mu})\mathrm{d}\xi \\
&\le \int_0^1 \left\|\nabla_{\boldsymbol{\mu}+\xi(\widehat{\boldsymbol{\mu}}^{(t)}-\boldsymbol{\mu})}^2\mathcal{L} - \nabla_{\boldsymbol{\mu}}^2\mathcal{L}\right\|\|\widehat{\boldsymbol{\mu}}^{(t)} - \boldsymbol{\mu}\|^2\mathrm{d}\xi \\
&\overset{(b)}{\le} \int_0^1 L\xi\|\widehat{\boldsymbol{\mu}}^{(t)} - \boldsymbol{\mu}\|^3\mathrm{d}\xi = L\|\widehat{\boldsymbol{\mu}}^{(t)} - \boldsymbol{\mu}\|^3,
\end{aligned}$$

where Inequality $(b)$ follows from the fact that $\nabla^2\mathcal{L}$ is twice continuously differentiable, its Jacobian is Lipchitz continuous in a neighborhood of $\boldsymbol{\mu}$, and $L$ is the Lipchitz constant.

Therefore, there exists a constant $\lambda > 0$ such that

$$\begin{aligned}
&\|\widehat{\boldsymbol{\mu}}^{(t)} - \boldsymbol{\mu}\|^2 + \Big\langle\widehat{\boldsymbol{\mu}}^{(t)} - \boldsymbol{\mu}, 2\eta^{(t)}\nabla_{\boldsymbol{\mu}}\mathcal{L}^{(t)}\Big\rangle \\
&\le \|\widehat{\boldsymbol{\mu}}^{(t)} - \boldsymbol{\mu}\|^2 + 2\eta^{(t)}(\widehat{\boldsymbol{\mu}}^{(t)} - \boldsymbol{\mu})^\top\nabla_{\boldsymbol{\mu}}^2\mathcal{L}(\widehat{\boldsymbol{\mu}}^{(t)} - \boldsymbol{\mu}) + 2\eta^{(t)}L\|\widehat{\boldsymbol{\mu}}^{(t)} - \boldsymbol{\mu}\|^3 \\
&\overset{(c)}{\le} \Big(1 - 2\eta^{(t)}\lambda\Big)\|\widehat{\boldsymbol{\mu}}^{(t)} - \boldsymbol{\mu}\|^2 + 2\eta^{(t)}L\|\widehat{\boldsymbol{\mu}}^{(t)} - \boldsymbol{\mu}\|^3,
\end{aligned} \tag{C.24}$$

where Inequality $(c)$ follows from Lemma 3 which proves $\nabla_{\boldsymbol{\mu}}^2\mathcal{L}$ is negative definite.

Meanwhile, Lemma 4 ensures with probability at least $1 - \exp(-\sqrt{M})$,

$$\eta^{(t)}\left\|\nabla\mathcal{L} - \nabla\widehat{\mathcal{L}}\right\| \le c_1\,\eta^{(t)}M^{-\frac{1}{4}}, \tag{C.25}$$

$$(\eta^{(t)})^2\left\|\nabla_{\widehat{\boldsymbol{\mu}}^{(t)}}\widehat{\mathcal{L}}\right\|^2 \le c_2(\eta^{(t)})^2 M^{-\frac{1}{2}} + c_3(\eta^{(t)})^2. \tag{C.26}$$

Substituting (C.24), (C.25), and (C.26) into (C.23) then yields the one-step error recursion

$$\|\widehat{\boldsymbol{\mu}}^{(t+1)} - \boldsymbol{\mu}\|^2 \leq \left(1 - 2\eta^{(t)}\lambda\right)\|\widehat{\boldsymbol{\mu}}^{(t)} - \boldsymbol{\mu}\|^2 + 2\eta^{(t)}L\|\widehat{\boldsymbol{\mu}}^{(t)} - \boldsymbol{\mu}\|^3$$
$$+ c_1\eta^{(t)}M^{-\frac{1}{4}} + c_2(\eta^{(t)})^2 M^{-\frac{1}{2}} + c_3(\eta^{(t)})^2. \tag{C.27}$$

Next, we aim prove $\|\widehat{\boldsymbol{\mu}}^{(t)} - \boldsymbol{\mu}\|^2 \leq K/(t + T')$ for a positive constant $K$ by induction. For some $p \geq 4$, let

$$\eta^{(t)} = \frac{\alpha}{t + T'},$$
$$M^{(t)} = (t + T')^p. \tag{C.28}$$

First, assume $\|\widehat{\boldsymbol{\mu}}^{(t)} - \boldsymbol{\mu}\|^2 \leq K/(t + T')$ for a fixed $t \geq 1$. From Equation (C.27), we note that there exists a constant $c_4 > 0$ such that

$$\|\widehat{\boldsymbol{\mu}}^{(t+1)} - \boldsymbol{\mu}\|^2 \leq \left(1 - 2\frac{\alpha\lambda}{t + T'}\right)\|\widehat{\boldsymbol{\mu}}^{(t)} - \boldsymbol{\mu}\|^2 + c_3\frac{\alpha^2}{(t + T')^2} + 2\frac{\alpha L}{t + T'}\|\widehat{\boldsymbol{\mu}}^{(t)} - \boldsymbol{\mu}\|^3 + c_4\frac{\alpha}{t + T'}t^{-\frac{p}{4}}$$

$$\leq \left(1 - 2\frac{\alpha\lambda}{t + T'}\right)\frac{K}{t + T'} + 2\frac{\alpha L}{t + T'}\left(\frac{K}{t + T'}\right)^{\frac{3}{2}} + c_4\alpha(t + T')^{-(1+\frac{p}{4})} + c_3\frac{\alpha^2}{(t + T')^2}.$$

Therefore, we have

$$\|\widehat{\boldsymbol{\mu}}^{(t+1)} - \boldsymbol{\mu}\|^2 - \frac{K}{t + T' + 1}$$

$$\leq \left(1 - 2\frac{\alpha\lambda}{t + T'}\right)\frac{K}{t + T'} + 2\frac{\alpha L}{t + T'}\left(\frac{K}{t + T'}\right)^{\frac{3}{2}} + c_4\alpha(t + T')^{-(1+\frac{p}{4})} + c_3\frac{\alpha^2}{(t + T')^2} - \frac{K}{t + T'} + \frac{K}{(t + T')^2}$$

$$= (-2\alpha\lambda + 1)\frac{K}{(t + T')^2} + 2\frac{\alpha L}{t + T'}\left(\frac{K}{t + T'}\right)^{\frac{3}{2}} + c_4\alpha(t + T')^{-(1+\frac{p}{4})} + c_3\frac{\alpha^2}{(t + T')^2}. \tag{C.29}$$

By choosing $\alpha \geq 1/\lambda$, $K \geq \max\{3c_3\alpha^2, 3c_4\alpha\}$ and $T' \geq 36\alpha^2 L^2 K$, we have

$$(-2\alpha\lambda + 1)\frac{K}{(t + T')^2} \leq -\frac{K}{(t + T')^2},$$
$$2\frac{\alpha L}{t + T'}\left(\frac{K}{t + T'}\right)^{\frac{3}{2}} \leq \frac{K}{3(t + T')^2},$$
$$c_4\alpha(t + T')^{-\left(1+\frac{p}{4}\right)} \leq \frac{K}{3(t + T')^2}, \tag{C.30}$$
$$c_3\frac{\alpha^2}{(t + T')^2} \leq \frac{K}{3(t + T')^2}.$$

Therefore, by substituting Equation (C.30) into Equation (C.29), we have

$$\|\widehat{\boldsymbol{\mu}}^{(t+1)} - \boldsymbol{\mu}\|^2 - \frac{K}{t + T' + 1} \leq 0,$$

which implies

$$\|\widehat{\boldsymbol{\mu}}^{(t+1)} - \boldsymbol{\mu}\|^2 \leq \frac{K}{t + T' + 1}$$
$$= \frac{K + 1}{t + T' + 1} \cdot \frac{K}{K + 1}$$
$$\leq \frac{K + 1}{t + T' + 1} \cdot \frac{t + T' + 1}{t + T' + \frac{T'}{K}}$$
$$= \frac{K + 1}{t + T' + \frac{T'}{K}}, \quad \forall t \geq 0.$$

Recall Lemma 1 indicates that, with probability at least $1 - \exp(-cNK/T')$, for some constant $c$, it holds that $\|\boldsymbol{\mu}^{(0)} - \boldsymbol{\mu}\| \leq K/T'$. Therefore, for any fixed $\epsilon \in [0, 1)$, let

$$N = \frac{T' \log(1/\epsilon)}{cK} \geq \frac{36\alpha^2 L^2}{c} \log(1/\epsilon), \tag{C.31}$$
$$M = (t + T')^4 \geq \max\{36\alpha^2 L^2 K, \log^2(1/\epsilon)\}.$$

Then, with probability at least $1 - \epsilon - e^{-\sqrt{M}}$, where the $e^{-\sqrt{M}}$ term arises from the condition required for Equation (C.21) to hold, the estimation error is upper bounded by

$$\|\widehat{\boldsymbol{\mu}}^{(t+1)} - \boldsymbol{\mu}\|^2 \leq \frac{K+1}{t + T' + \frac{T'}{K}}$$
$$\leq \frac{K+1}{\sqrt[4]{M} + \frac{Nc}{\log(1/\epsilon)}}. \tag{C.32}$$

For sufficiently small $\epsilon$ such that $\log(1/\epsilon) \geq c$, denoting $c' = \frac{K+1}{c}$, we obtain

$$\frac{K+1}{\sqrt[4]{M} + \frac{Nc}{\log(1/\epsilon)}} \leq \frac{(K+1)\log(1/\epsilon)}{\min\{\log(1/\epsilon), c)(\sqrt[4]{M} + N)\}}$$
$$\leq c' \frac{\log(1/\epsilon)}{\sqrt[4]{M} + N}.$$

Since we let $\sqrt{M} = (T' + t)^2 \geq \log(1/\epsilon)$, we conclude that there exist a constant $c''$ such that with probability $1 - \epsilon$, the estimation error is upper bounded by

$$\|\widehat{\boldsymbol{\mu}}^{(t)} - \boldsymbol{\mu}\|^2 \leq c'' \frac{\log(1/\epsilon)}{N + \sqrt[4]{M}}.$$

This completes the proof of Theorem 4.2.

## C.3 Proof of Corollary 4.1

First, we restate the corollary below.

**Corollary C.1** (Restatement of Corollary 4.1). *Let $\widehat{\mathbf{y}}_j$ be the predicted label for $\mathbf{x}_j$ according to Equation (3.5). Let $\mathcal{R}^*$ be the prediction error under the Bayes-optimal classifier with known class mean vectors $\boldsymbol{\mu}_1, \cdots, \boldsymbol{\mu}_C$. Then, under the same conditions as described in Theorem 4.2, we have*

$$\mathbb{P}[\widehat{\mathbf{y}}_j \neq \mathbf{y}|\boldsymbol{\mu}_1, \cdots, \boldsymbol{\mu}_C] - \mathcal{R}^* \leq \mathcal{O}(1/\sqrt{N + \text{poly}(M)}).$$

*Proof.* First, we define $\Delta = \|\widehat{\mathbf{M}} - \mathbf{M}\|_F$, define $\widehat{g}$ as the Bayes-optimal classifier given estimated class means $\widehat{\mathbf{M}}$ and define $g$ as the Bayes-optimal classifier given ground truth class means $\mathbf{M}$. Suppose $\widehat{g}(\mathbf{x}) \neq g(\mathbf{x})$. Then, there exist indices $i \neq k$ such that $g(\mathbf{x}) = i$ and $\widehat{g}(\mathbf{x}) = k$. Because $g(\mathbf{x}) = i$ is Bayes-optimal, we have

$$\|\mathbf{x} - \boldsymbol{\mu}_i\| \leq \|\mathbf{x} - \boldsymbol{\mu}_k\| \text{ and } \|\mathbf{x} - \widehat{\boldsymbol{\mu}}_k\| \leq \|\mathbf{x} - \widehat{\boldsymbol{\mu}}_i\|.$$

Denote $\zeta = \|\boldsymbol{\mu}_i - \boldsymbol{\mu}_k\|$. Therefore, from the geometric observation, the misclassification only happens when $\mathbf{x}$ is in the dihedral cone with angle $\theta$, where $\tan(\theta) = \Delta/\zeta$ (Diakonikolas et al., 2018).Thus, the probability for misclassification is upper bounded

$$\mathbb{P}[\widehat{g}(\mathbf{x}) \neq g(\mathbf{x})] \leq c'\theta,$$

for a positive constant $c'$. Since $\mathbb{P}[\widehat{\mathbf{y}}_j \neq \mathbf{y}|\boldsymbol{\mu}_1, \cdots, \boldsymbol{\mu}_C] - \mathcal{R}^* = \mathbb{P}[\widehat{g}(\mathbf{x}) \neq g(\mathbf{x})]$ and from Theorem 4.2 we have $\Delta \leq c'\sqrt{1/(N + \sqrt[4]{M})}$ for positive constant $c'$, the proof is thus complete. $\square$

# D Proof of Training Dynamics

First, we restate Theorem 5.1 below.

**Theorem D.1** (Restatement of Theorem 5.1). *Let* $\{\mathbf{Q}^{(k)}, \mathbf{K}^{(k)}, \mathbf{V}^{(k)}\}_{k \geq 0}$ *be the parameters of the first attention layer of the transformer after applying $k$ iterations of gradient descent on the population loss defined in Equation (5.2). Then, with the initialization specified in Assumption 1, we have*

$$\|\mathbf{W}^{(k)} - \mathbf{\Sigma}^{-1}\|_F^2 \leq c^k \|\mathbf{W}^{(0)} - \mathbf{\Sigma}^{-1}\|_F^2,$$

*for some positive constant $c$, while the other parameters in $\mathbf{Q}^{(0)}$, $\mathbf{K}^{(0)}$ and $\mathbf{V}^{(0)}$ remain unchanged.*

We assume ground truth means are IID sampled from standard Gaussian distribution: $\boldsymbol{\mu}_i \sim \mathcal{N}(\mathbf{0}, \mathbf{I})$ for all $i$. Then, we introduce the following quantities: 1) the formulation of class mean estimations given by the transformer during teacher forcing training; 2) the reference class mean estimations given by the reference policy; and 3) the formulation of the gradient of the teacher forcing training loss.

At the $k$-th GD iteration during training, we denote the set of reference class mean estimations as $\boldsymbol{\mu}_{\text{ref},1}^{(k,t)}, \cdots, \boldsymbol{\mu}_{\text{ref},C}^{(k,t)}$ for the CoT steps $t \in [T]$. Given the reference class mean estimations, the estimation given by the transformer throughout teacher forcing satisfies

$$\widehat{\boldsymbol{\mu}}_i^{(k,t+1)} = \boldsymbol{\mu}_{\text{ref},i}^{(k,t)} - \frac{\eta^{(t)}}{M} \sum_{j=N+1}^{N+M} \widehat{p}_{ij}^{(k,t)} \left( \boldsymbol{\mu}_{\text{ref},i}^{(k,t)} - \mathbf{x}_j \right),$$

where $\widehat{p}_{ij}^{(k,t)}$ is given by

$$\widehat{p}_{ij}^{(k,t)} = \frac{\sum_{\tau=0}^{t} \exp\left( -\frac{w}{2} \|\widehat{\boldsymbol{\mu}}_i^{(\tau)}\|^2 + \mathbf{x}_j^{\top} \mathbf{W}^{(k)} \widehat{\boldsymbol{\mu}}_i^{(\tau)} + \beta\tau \right)}{\sum_{\tau=0}^{t} \sum_{c=1}^{C} \exp\left( -\frac{w}{2} \|\widehat{\boldsymbol{\mu}}_c^{(\tau)}\|^2 + \mathbf{x}_j^{\top} \mathbf{W}^{(k)} \widehat{\boldsymbol{\mu}}_c^{(\tau)} + \beta\tau \right)}.$$

By choosing $\beta \to \infty$, we further have

$$\widehat{p}_{ij}^{(k,t)} = \frac{\exp\left( -\frac{w}{2} \|\widehat{\boldsymbol{\mu}}_i^{(\tau)}\|^2 + \mathbf{x}_j^{\top} \mathbf{W}^{(k)} \widehat{\boldsymbol{\mu}}_i^{(\tau)} \right)}{\sum_{c=1}^{C} \exp\left( -\frac{w}{2} \|\widehat{\boldsymbol{\mu}}_c^{(\tau)}\|^2 + \mathbf{x}_j^{\top} \mathbf{W}^{(k)} \widehat{\boldsymbol{\mu}}_c^{(\tau)} \right)}.$$

We choose the reference policy under which

$$\boldsymbol{\mu}_{\text{ref},i}^{(k,t+1)} = \boldsymbol{\mu}_{\text{ref},i}^{(k,t)} - \frac{\eta^{(t)}}{M} \sum_{j=N+1}^{N+M} p_{ij}^{(k,t)} \left( \boldsymbol{\mu}_{\text{ref},i}^{(k,t)} - \mathbf{x}_j \right),$$

with

$$p_{ij}^{(k,t)} = \frac{\exp\left( -\frac{1}{2} \|\boldsymbol{\mu}_{\text{ref},i}^{(k,t)} - \mathbf{x}_j\|_{\Sigma^{-1}}^2 \right)}{\sum_{c=1}^{C} \exp\left( -\frac{1}{2} \|\boldsymbol{\mu}_{\text{ref},c}^{(k,t)} - \mathbf{x}_j\|_{\Sigma^{-1}}^2 \right)}.$$

To simplify the notation, when there is no ambiguity, we drop the superscript $(k)$ for the training iteration. Denote $\widehat{\mathbf{q}}_j^{(t)} = [\widehat{p}_{1j}^{(t)} \cdots \widehat{p}_{Cj}^{(t)}]$ and $\mathbf{q}_j^{(t)} = [p_{1j}^{(t)} \cdots p_{Cj}^{(t)}]$. At the $k$-th training iteration, the CoT training loss with teacher forcing is

$$\widehat{\mathcal{L}}_{\text{CoT-train}}(\boldsymbol{\Theta}; \mathcal{I}_{\mathbf{M}}) = \frac{1}{T} \sum_{t=1}^{T} \sum_{j=N+1}^{N+M} \text{CE}\left( \mathbf{q}_j^{(t)}, [\text{TF}_{\boldsymbol{\Theta}}(\mathbf{H}_{\text{ref}}^{(t-1)})]_{2d+2c+1:2d+3c, N+j} \right)$$

$$= \frac{1}{T} \sum_{t=1}^{T} \sum_{j=N+1}^{N+M} \text{CE}\left( \mathbf{q}_j^{(t)}, \widehat{\mathbf{q}}_j^{(t)} \right),$$

where CE is the cross entropy loss function.

Define $s_{ij}^{(t)} = -\frac{1}{2}\|\widehat{\boldsymbol{\mu}}_i^{(\tau)}\|^2 + \mathbf{x}_j^\top \mathbf{W}^{(k)}\widehat{\boldsymbol{\mu}}_i^{(\tau)}$ and $\mathbf{s}_j^{(t)} = [s_{1j}^{(t)}\cdots s_{Cj}^{(t)}]$. Note that the derivative can be written as

$$\frac{\partial \mathrm{CE}\left(\mathbf{q}_j^{(t)}, \widehat{\mathbf{q}}_j^{(t)}\right)}{\partial s_{ij}^{(t)}} = \frac{\partial \frac{\exp(s_{ij}^{(t)})}{\sum_{k=1}^C \exp(s_{kj}^{(t)})}}{\partial s_{ij}^{(t)}} = \widehat{p}_{ij}^{(t)} - p_{ij}^{(t)}.$$

Furthermore, since $\partial s_{ij}^{(t)}/\partial \mathbf{W}_{ab} = \mathbf{M}_{a,i}\mathbf{x}_{jb}$, where $a, b \in [d]$, by the chain rule, we have

$$\frac{\partial \mathrm{CE}\left(\mathbf{q}_j^{(t)}, \widehat{\mathbf{q}}_j^{(t)}\right)}{\partial \mathbf{W}_{ab}} = \frac{\partial \mathrm{CE}\left(\mathbf{q}_j^{(t)}, \widehat{\mathbf{q}}_j^{(t)}\right)}{\partial s_{ij}^{(t)}} \frac{\partial s_{ij}^{(t)}}{\partial \mathbf{W}_{ab}} = \sum_i (\widehat{p}_{ij}^{(t)} - p_{ij}^{(t)})\mathbf{M}_{a,i}\mathbf{x}_{jb}. \tag{D.1}$$

Based on the notations, we will prove Theorem 5.1 as follows.

**Step 1: Given the gradient of the cross entropy loss with respect to the learnable parameter matrix W, our first step is to provide a decomposition of the gradient so that it becomes analytically tractable.**

In the matrix form, Equation (D.1) can be written as

$$\nabla_{\mathbf{W}}\mathrm{CE}\left(\mathbf{q}_j^{(t)}, \widehat{\mathbf{q}}_j^{(t)}\right) = \mathbf{M}(\widehat{\mathbf{q}}_j^{(t)} - \mathbf{q}_j^{(t)})\mathbf{x}_j^T.$$

By the Stein's lemma, we have

$$\mathbb{E}[\mathbf{M}(\widehat{\mathbf{q}}_j^{(t)} - \mathbf{q}_j^{(t)})\mathbf{x}_j^\top]$$
$$= \mathbb{E}_{\mathbf{M}}\left[\mathbf{M}\left[\mathbb{E}_{\mathbf{x}_j}[\widehat{\mathbf{q}}_j^{(t)} - \mathbf{q}_j^{(t)}]\mathbb{E}[\mathbf{x}_j^\top] + \mathbb{E}_{\mathbf{x}_j}[\nabla\widehat{\mathbf{q}}_j^{(t)} - \nabla\mathbf{q}_j^{(t)}]\boldsymbol{\Sigma}\right]\right]$$
$$= \underbrace{\mathbb{E}_{\mathbf{M}}\left[\mathbf{M}\mathbb{E}_{\mathbf{x}_j}[\widehat{\mathbf{q}}_j^{(t)} - \mathbf{q}_j^{(t)}]\mathbb{E}[\mathbf{x}^\top]\right]}_{\mathcal{A}_1} + \underbrace{\mathbb{E}_{\mathbf{M}}\left[\mathbf{M}\mathbb{E}_{\mathbf{x}_j}[\nabla\widehat{\mathbf{q}}_j^{(t)} - \nabla\mathbf{q}_j^{(t)}]\boldsymbol{\Sigma}\right]}_{\mathcal{A}_2}.$$

**Step 2: Based on the decomposition, we aim to show that $\mathcal{A}_1 = 0$.**

We note that when taking the expectation over the labeled dataset, we have $\mathbb{E}\left[\frac{C}{N}\sum_{j\in[N]}\mathbf{x}_j \cdot \left(\mathbf{e}_i^\top\mathbf{y}_j\right)\right] = \boldsymbol{\mu}_i$. Therefore, $\boldsymbol{\mu}_{\mathrm{ref,i}}^0 = \boldsymbol{\mu}_i$. When the reference class mean estimations are generated by gradient descent over the population loss, we have $\boldsymbol{\mu}_{\mathrm{ref},i}^{(t)} = \boldsymbol{\mu}_i$ for any $i \in [C]$ and $t \in [T]$ the gradient over the population loss is zero:

$$\mathbb{E}_{\mathbf{x}}\left[\frac{\exp\left(-\frac{1}{2}\|\boldsymbol{\mu}_{\mathrm{ref},i}^{(t)} - \mathbf{x}_j\|_{\boldsymbol{\Sigma}^{-1}}^2\right)}{\sum_{c=1}^C \exp\left(-\frac{1}{2}\|\boldsymbol{\mu}_{\mathrm{ref},c}^{(t)} - \mathbf{x}_j\|_{\boldsymbol{\Sigma}^{-1}}^2\right)}\left(\boldsymbol{\mu}_{\mathrm{ref},i}^{(t)} - \mathbf{x}_j\right)\right]$$
$$= \int_{\mathbb{R}^d}\frac{\exp(-\frac{1}{2}\|\boldsymbol{\mu}_{\mathrm{ref},i}^{(t)} - \mathbf{x}\|_{\boldsymbol{\Sigma}^{-1}}^2)}{\sum_{c=1}^C \exp(-\frac{1}{2}\|\boldsymbol{\mu}_{\mathrm{ref},c}^{(t)} - \mathbf{x}\|_{\boldsymbol{\Sigma}^{-1}}^2)}\left[\sum_{k=1}^C \frac{1}{C}\varphi_k(\mathbf{x})\right](\boldsymbol{\mu}_{\mathrm{ref},i}^{(t)} - \mathbf{x})d\mathbf{x},$$
$$\stackrel{(a)}{=} \int_{\mathbb{R}^d}\frac{1}{C}\varphi_i(\mathbf{x})(\boldsymbol{\mu}_i - \mathbf{x})d\mathbf{x} = 0,$$

where $\varphi_i(\mathbf{x})$ is the pdf of Gaussian distribution with mean $\boldsymbol{\mu}_i$ and covariance matrix $\boldsymbol{\Sigma}$, and equality $(a)$ holds since $\boldsymbol{\mu}_{\mathrm{ref},i}^{(k,t)} = \boldsymbol{\mu}_i$. Given the above-discussed property of the reference class mean estimations, for $\mathbb{E}_{\mathbf{x}_j}[\widehat{\mathbf{q}}_j^{(t)} - \mathbf{q}_j^{(t)}]$ in $\mathcal{A}_1$, its is obvious that $\mathbb{E}_{\mathbf{x}_j}[\mathbf{q}_j^{(t)}] = 1/C$. For $\mathbb{E}_{\mathbf{x}_j}[\widehat{\mathbf{q}}_j^{(t)}]$, we let $\mathbf{W}^{(0)}$ initialize form a isotropic matrix $w\mathbf{I}$, and we assume at training iteration step $k$, it preserve the isotropic as $\mathbf{W}^{(k)}$. Therefore, since the ground truth $\boldsymbol{\Sigma}$ is an isotropic matrix, the temperature acts identically on all classes:

$$\mathbb{E}\left[\frac{\exp\left(-\frac{\alpha}{2}\|\widehat{\boldsymbol{\mu}}_i^{(\tau)} - \mathbf{x}_j\|^2\right)}{\sum_{c=1}^C \exp\left(-\frac{\alpha}{2}\|\boldsymbol{\mu}_c - \mathbf{x}_j\|^2\right)}\right] = \mathbb{E}\left[\frac{\exp\left(-\frac{1}{2}\|\widehat{\boldsymbol{\mu}}_i^{(\tau)} - \mathbf{x}_j\|_{\boldsymbol{\Sigma}^{-1}}^2\right)}{\sum_{c=1}^C \exp\left(-\frac{1}{2}\|\boldsymbol{\mu}_c - \mathbf{x}_j\|_{\boldsymbol{\Sigma}^{-1}}^2\right)}\right].$$

Therefore, we have $\mathbb{E}_{\mathbf{x}_j}[\widehat{\mathbf{p}}_j^{(t)} - \mathbf{p}_j^{(t)}] = 0$, which gives $\mathcal{A}_1 = 0$.

**Step 3: Finally, we analyze the properties of $\mathcal{A}_2$, and obtain the final results afterwards.** We will prove that $\mathbf{W}^{(t)}$ preserves isotropic by induction. Note that we assume training iteration step $k$, $\mathbf{W}^{(k)}$ is isotropic. Besides, we initialize $\mathbf{W}^{(0)}$ as an isotropic matrix.

$\mathcal{A}_2$ can be rewritten as

$$
\begin{aligned}
\mathcal{A}_2 &= \mathbb{E}_{\mathbf{M}} \left[ \mathbf{M} \mathbb{E}_{\mathbf{x}_j} [\nabla \widehat{\mathbf{q}}_j^{(t)} - \nabla \mathbf{q}_j^{(t)}] \mathbf{\Sigma} \right] \\
&= \mathbb{E}_{\mathbf{M}} \left[ \mathbf{M} \left( \left( \mathrm{diag}(\mathbb{E}[\widehat{\mathbf{q}}_j^{(t)}]) - \mathbb{E}_{\mathbf{x}}[\widehat{\mathbf{q}}_j^{(t)}(\widehat{\mathbf{q}}_j^{(t)})^\top] \right) \mathbf{M}^\top \mathbf{W}^{(k)} \mathbf{\Sigma} - \left( \mathrm{diag}(\mathbb{E}[\mathbf{q}_j^{(t)}]) - \mathbb{E}_{\mathbf{x}}[\mathbf{q}_j^{(t)}(\mathbf{q}_j^{(t)})^\top] \right) \mathbf{M}^\top \right) \right].
\end{aligned}
$$

Because the class prior is uniform and the isotropic initialisation, we have $\mathbb{E}_{\mathbf{x}_j}[\widehat{\mathbf{q}}_j^{(t)}] = \mathbb{E}_{\mathbf{x}_j}[\mathbf{q}_j^{(t)}] = 1/C$. Since each coordinate of $\widehat{\mathbf{q}}_j^{(t)}$ (or $\mathbf{q}_j^{(t)}$) has the same marginal distribution and any two distinct coordinates have the same joint distribution, we have

$$
\mathrm{diag}(\mathbb{E}[\widehat{\mathbf{q}}_j^{(t)}]) = \mathrm{diag}(\mathbf{q}_j^{(t)}) = \frac{1}{C}\mathbf{I}, \qquad \mathbb{E}_{\mathbf{x}}[\widehat{\mathbf{q}}_j^{(t)}(\widehat{\mathbf{q}}_j^{(t)})^\top] = \mathbb{E}_{\mathbf{x}}[\mathbf{q}_j^{(t)}(\mathbf{q}_j^{(t)})^\top] = \frac{1}{C^2}\mathbf{1}\mathbf{1}^\top.
$$

Therefore, we have

$$
\mathcal{A}_2 = \mathbb{E}_{\mathbf{M}} \left[ \mathbf{M} \left( \mathrm{diag}(1/C) - \frac{1}{C^2}\mathbf{1}\mathbf{1}^\top \right) \mathbf{M}^\top \left( \mathbf{W}^{(k)}\mathbf{\Sigma} - \mathbf{I} \right) \right].
$$

Note that $\nabla_{\mathbf{W}} L_{\mathrm{CoT}}(\mathbf{W}^{(k)}) = \mathcal{A}_2$, therefore, we obtain

$$
\begin{aligned}
\|\nabla_{\mathbf{W}} L_{\mathrm{CoT}}(\mathbf{W}^{(k)})\|_F &= \left\| \mathbb{E}_{\mathbf{M}} \left[ \mathbf{M} \left( \mathrm{diag}(1/C) - \frac{1}{C^2}\mathbf{1}\mathbf{1}^\top \right) \mathbf{M}^\top \left( \mathbf{W}^{(k)}\mathbf{\Sigma} - \mathbf{I} \right) \right] \right\|_F \\
&= \left\| \mathbb{E}_{\mathbf{M}} \left[ \frac{1}{C}\mathbf{M}\mathbf{M}^\top - \frac{1}{C^2}\mathbf{M}\mathbf{1}\mathbf{1}^\top\mathbf{M}^\top \right] \left( \mathbf{W}^{(k)}\mathbf{\Sigma} - \mathbf{I} \right) \right\|_F \\
&= \sigma^2 \left( 1 - \frac{1}{C} \right) \left\| \left( \mathbf{W}^{(k)} - \mathbf{\Sigma}^{-1} \right) \right\|_F.
\end{aligned}
$$

Since all columns in $\mathbf{M}$ are sampled from $\mathcal{N}(\mathbf{0}, \mathbf{I})$ and $\mathbf{W}^{(k)}$ is assumed to be an isotropic matrix, it's obvious that $\mathcal{A}_2$ is also an isotropic matrix. It follows that

$$
\begin{aligned}
&\langle \mathbf{W}^{(k)} - \mathbf{\Sigma}^{-1}, \nabla_{\mathbf{W}} L_{\mathrm{CoT}} \rangle \\
&= \mathbb{E}_{\mathbf{M}} \left[ \mathrm{trace} \left( \mathbf{M} \left( \mathrm{diag}(1/C) - \frac{1}{C^2}\mathbf{1}\mathbf{1}^\top \right) \mathbf{M}^\top \left( \mathbf{W}^{(k)} - \mathbf{\Sigma}^{-1} \right) \mathbf{\Sigma} \left( \mathbf{W}^{(k)} - \mathbf{\Sigma}^{-1} \right)^\top \right) \right] \\
&\overset{(a)}{=} \sigma^2 \mathrm{trace} \left( \mathbb{E}_{\mathbf{M}} \left[ \frac{1}{C}\mathbf{M}\mathbf{M}^\top \right] \left( \mathbf{W}^{(k)} - \mathbf{\Sigma}^{-1} \right) \left( \mathbf{W}^{(k)} - \mathbf{\Sigma}^{-1} \right)^\top \right. \\
&\qquad \left. - \mathbb{E}_{\mathbf{M}} \left[ \frac{1}{C^2}\mathbf{M}\mathbf{1}\mathbf{1}^\top\mathbf{M}^\top \right] \left( \mathbf{W}^{(k)} - \mathbf{\Sigma}^{-1} \right) \left( \mathbf{W}^{(k)} - \mathbf{\Sigma}^{-1} \right)^\top \right) \\
&= \sigma^2 \mathrm{trace} \left( \left( \mathbf{W}^{(k)} - \mathbf{\Sigma}^{-1} \right) \left( \mathbf{W}^{(k)} - \mathbf{\Sigma}^{-1} \right)^\top - \frac{1}{C} \left( \mathbf{W}^{(k)} - \mathbf{\Sigma}^{-1} \right) \left( \mathbf{W}^{(k)} - \mathbf{\Sigma}^{-1} \right)^\top \right) \\
&= \sigma^2 \left( 1 - \frac{1}{C} \right) \|\mathbf{W}^{(k)} - \mathbf{\Sigma}^{-1}\|_F^2.
\end{aligned}
$$

where equation $(a)$ follows from the assumption that $\mathbf{\Sigma} = \sigma^2 \mathbf{I}$.

Let $\gamma = \sigma^2(1 - 1/C)$. Then,

$$
\begin{aligned}
\|\mathbf{W}^{(k+1)} - \mathbf{\Sigma}^{-1}\|_F^2 &\leq \|\mathbf{W}^{(k)} - \mathbf{\Sigma}^{-1}\|_F^2 - 2\gamma\eta\|\mathbf{W}^{(k)} - \mathbf{\Sigma}^{-1}\|_F^2 + \eta^2\gamma^2\|\mathbf{W}^{(k)} - \mathbf{\Sigma}^{-1}\|_F^2 \\
&\leq (1 - \gamma\eta)^2 \|\mathbf{W}^{(k)} - \mathbf{\Sigma}^{-1}\|_F^2
\end{aligned}
$$

Select step size such that $(1 - \gamma\eta)^2 \leq 1$ and let $c := (1 - \gamma\eta)^2$, we obtain

$$
\|\mathbf{W}^{(k)} - \mathbf{\Sigma}^{-1}\|_F^2 \leq c^k \|\mathbf{W}^{(0)} - \mathbf{\Sigma}^{-1}\|_F^2.
$$

# E  Auxiliary Lemmas

**Lemma 5** (Stein's Lemma). *Let $X \in \mathbb{R}^d$ be a random vector with*

$$X \sim \mathcal{N}(\mu, \Sigma),$$

*where $\mu \in \mathbb{R}^d$ and $\Sigma \in \mathbb{R}^{d \times d}$ is a positive definite matrix. Let $f : \mathbb{R}^d \to \mathbb{R}^k$ be a continuously differentiable function such that*

$$\mathbb{E}\big[\|f(X)\|\big] < \infty \quad and \quad \mathbb{E}\big[\|\nabla f(X)\|_F\big] < \infty,$$

*where $\|\cdot\|$ denotes the Euclidean norm in $\mathbb{R}^k$ and $\|\cdot\|_F$ is the Frobenius norm. Then, the following identity holds:*

$$\mathbb{E}\Big[(X - \mu)\, f(X)^T\Big] = \Sigma\, \mathbb{E}\Big[\nabla f(X)\Big],$$

*where $\nabla f(X)$ is the $k \times d$ Jacobian matrix of $f$ evaluated at $X$.*

