# OpenReview forum: "Unlabeled Data Can Provably Enhance In-Context Learning of Transformers"
_NeurIPS.cc/2025/Conference — NeurIPS 2025 poster_

### Official Review · Reviewer_uxcu · 2025-06-27

**Clarity:** 2
**Significance:** 2
**Originality:** 2
**Rating:** 4
**Confidence:** 3

**Summary:**

This paper proposes a novel augmented ICL framework, where the prompt contains some labeled examples as well as unlabeled inputs. This framework is motivated by scarcity of labeled data and high cost to get labels for the readily available unlabeled data. Considering the setting of multi-class linear classification, the paper first presents a construction showing that the transformer output (class mean estimates) converges to the ground truth as the number of chain-of-thought (CoT) steps is increased. This is accomplished by obtaining initial estimates from the labeled examples and refining them using the unlabeled inputs emulating the expectation-maximization (EM) algorithm. This leads to an excess risk bound of $O(1/\sqrt{N \text{poly}(M)})$. Next, it shows that such a transformer can be trained by teacher forcing using a specific initialization. The theoretical result is supported by numerical experiments.

**Questions:**

Other suggestions about the paper writing:

- There are several typos which should be corrected, e.g., it should be ‘simplify’ in line 73, ‘transformers’ in line 93, ‘exists’ in line 193, ‘taking’ in line 221, ‘\mathbf{\mu}’ in line 232, ‘combining’ in line 248, to list some of them.
- The proof sketches are helpful, but the initial discussion in lines 72-75 in the Introduction is not very clear and can be refined.

**Ethical Concerns:**

["NO or VERY MINOR ethics concerns only"]

**Final Justification:**

The rebuttal addressed my concerns, so I increased my score from 3 to 4.

**Limitations:**

yes

**Quality:**

3

**Strengths And Weaknesses:**

The main strengths of the paper are:

- It proposes a novel augmented ICL framework (apart from the concurrent work [1]).
- It is the first theoretical study showing the benefit of unlabeled data for ICL in transformers.

However, the paper has some important weaknesses that should be addressed.

- As the paper mainly focuses on theory, it should clearly discuss the novelty or relation with prior work for the proof techniques for the main results. Specifically, this prior work [2] comparing transformers with EM for multi-class clustering seems quite relevant and should be discussed.
- It is unclear how effective the proposed augmented ICL framework can be in practice. I strongly suggest that the paper should include experimental results for at least one setting when prompting a pretrained LLM with this framework.
- The paper is missing references to several relevant related works on understanding ICL in transformers in Section 2, e.g., [3-6] to list a few. Only four papers are discussed, which does not include the seminal work Garg et al., 2022, which is strange. More relevant papers should be mentioned/discussed. Also, a minor comment about line 29: Fu et al., 2023 is not a seminal work on ICL in transformers and can be directly cited in Section 2 rather than the introduction.

References:

[1] Li et al., “When and How Unlabeled Data Provably Improve In-Context Learning”, 2025.

[2] He et al., “Transformers versus the EM Algorithm in Multi-class Clustering”, 2025.

[3] von Oswald et al., “Transformers learn in-context by gradient descent”, ICML 2023.

[4] Akyurek et al., “What learning algorithm is in-context learning? investigations with linear models”, ICLR 2023.

[5] Xie et al., “An explanation of in-context learning as implicit bayesian inference”, ICLR 2022.

[6] Raventos et al., “Pretraining task diversity and the emergence of non-bayesian in-context learning for regression”, NeurIPS 2023.

---

> ### Author Rebuttal · Authors · 2025-07-31
>
> We thank the reviewer for the careful reading and thoughtful comments. Please see our responses below.
>
>
> ***Weakness 1:*** Lack of discussion of the novelty or relation with prior work for the proof techniques for the main results.
>
> ***Response to Weakness 1:*** We appreciate the reviewer's insightful comment and for highlighting these highly relevant works. We clarify our theoretical contributions compared with [1] and [2] as follows.
>
> Our work significantly differs from [1] in both problem setting and theoretical guarantees/proof novelty. [1] primarily focuses on *unsupervised learning* of Gaussian mixture models, aiming to mimic a clustering algorithm (Lloyd's algorithm). Its analysis centers on showing that a transformer's output can approximate an iterative algorithm, which is susceptible to local optima, since for such a problem, the EM algorithm is prone to convergence at local optima and is highly sensitive to initialization. In contrast, our work addresses **semi-supervised learning** during ICL, where a small amount of labeled data is available. Our goal is to demonstrate how unlabeled data can effectively augment traditional ICL that typically relies solely on labeled data. This fundamental difference in setting leads to **distinct theoretical contributions**. Our key technical contribution is proving a non-asymptotic convergence to the *ground-truth means*, not just an algorithm's output. Our novel two-stage proof technique first shows how the transformer can leverage labeled data to learn a high-quality initialization for the means. Next, we prove that with this strong initialization, the transformer's EM-style updates on the unlabeled data lead to monotonous improvement and convergence to the true parameters. This method of using labeled data to bypass the local optima problem, which is common in unsupervised EM, is central to our theoretical framework.
>
> Compared with the concurrent work [2], which also aims to show that transformers can leverage unlabeled data in ICL, our theoretical contributions and technical analysis differ in several aspects:
>
> **Theoretical contributions:** **First**, [2] studies a linear transformer without non‑linear activations, whereas we analyze a more realistic architecture that includes the soft‑max attention mechanism. **Second**, their theory is restricted to a Gaussian‑mixture model with exactly two classes. Our framework allows an arbitrary (finite) number of classes, covering multi‑class scenarios. **Third**, Theorem 3 in [2] provides only an asymptotic guarantee as the number of unlabeled samples tends to infinity, and their finite‑sample bound includes a non‑vanishing bias term. By contrast, in our work, **we establish a rigorous, non-asymptotic convergence guarantee for a transformer with softmax non-linearity in a general multi-class setting**.
>
> **Technical novelty:** Our technical proofs rely on several novel ingredients. First, we formally characterize the initial condition required to show the non-asymptotic convergence. Then, and most critically, we tackle the challenge of analyzing the gradient dynamics over the unlabeled data. **A key innovation in our analysis is the strategic design of the temperature parameter $\beta\tau$.** We show how it can be set to ensure the model's attention weights are sharply focused on the most recent class mean estimate, effectively isolating the gradient updates at each step. By combining this with Bernstein's inequality, we derive a tight bound on the deviation between the true stochastic gradient and the gradient of the ideal expected log-likelihood. Finally, by leveraging the Lipschitz continuity of the loss landscape and integrating our novel gradient bound, we prove the finite-sample convergence of our model.
>
> Our analysis differs fundamentally from that of [2] in several ways. First, we analyze a multi-layer transformer with standard softmax attention, a more complex and general setup. In contrast, the theory in [2] is restricted to transformers with the simpler linear attention mechanism. Second, our construction enables the transformer to perform EM-style updates on the unlabeled data. The model in [2], however, is shown to implement a different learning rule: a semi-supervised plug-in estimator. As a result of these differences, our analysis establishes a more promising convergence property for the model compared to the results shown in [2].
>
> We'll incorporate this detailed comparison into the revised manuscript to clarify our theoretical contributions.
>
>
> ***Weakness 2:*** It is unclear how effective the proposed augmented ICL framework can be in practice.
>
> ***Response to Weakness 2:*** We have performed an additional experiment on a real-world dataset to verify the impact of unlabeled samples on the ICL performance of a pre-trained LLM. Please refer to our responses to Reviewer RyNA (Response to Weakness 1) for the experimental setup and results.
>
>
> ***Weakness 3:*** Missing references to several relevant related works in Section 2.
>
> ***Response to Weakness 3:*** We thank the reviewer for pointing out these important related works.
>
> We will thoroughly revise Section 2 based on your suggestion. We will add and discuss the seminal work by Garg et al. (2022), the additional references [1-6], and other relevant papers. Furthermore, we will move the citation for Fu et al. (2023) from the introduction to the related works section as advised.
>
>
>
> ***Question 1:*** Typos need to be fixed.
>
> ***Response to Question 1:***  We thank the reviewer for their careful reading and for pointing out these typos. We will correct all the typos mentioned and perform a thorough proofreading to eliminate any remaining typos.
>
> ***Question 2:*** Initial discussion in lines 72-75 is unclear.
>
>
> ***Response to Question 2:*** We appreciate the reviewer’s helpful comment. We will revise the parts in the introduction you mentioned as follows:
>
> *Our proof introduces a novel decomposition of the gradient of the CoT training loss by applying Stein's lemma, separating it into two analytically tractable components: a posterior-difference term and a Jacobian-difference term. Leveraging the inherent isotropy preserved during gradient descent, we show that the posterior-difference term vanishes, while symmetry properties simplify the Jacobian-difference term, enabling us to derive a tight upper bound on the key inner-product expression and establish the desired sublinear convergence rate.*
>
> References:
>
> [1] He et al., "Transformers versus the EM Algorithm in Multi-class Clustering", 2025.
>
> [2] Li et al., "When and How Unlabeled Data Provably Improve In-Context Learning", 2025.
>
> [3] von Oswald et al., “Transformers learn in-context by gradient descent”, ICML 2023.
>
> [4] Akyurek et al., “What learning algorithm is in-context learning? investigations with linear models”, ICLR 2023.
>
> [5] Xie et al., “An explanation of in-context learning as implicit Bayesian inference”, ICLR 2022.
>
> [6] Raventos et al., “Pretraining task diversity and the emergence of non-bayesian in-context learning for regression”, NeurIPS 2023.
>
> ----------
>
> We hope our responses have addressed your concerns satisfactorily. We sincerely appreciate your constructive feedback and would be grateful if you would consider raising your evaluation based on our responses. We would also be happy to address any further questions you may have.

---

> > ### Comment · Reviewer_uxcu · 2025-08-04
> >
> > Thank you. The rebuttal has addressed my concerns and I will increase my score to 4.

---

> > > ### Author Response · Authors · 2025-08-04
> > > **Thank you for your response**
> > >
> > > Dear reviewer uxcu:
> > >
> > > We are glad to know that our responses have addressed your concerns, and we truly appreciate that you recognize the contributions of this work and have increased your rating! Your thoughtful comments have helped us greatly improve the quality of this work, and we will definitely keep polishing the paper and incorporating your valuable suggestions into the final version.
> > >
> > > Thank you again for the insightful comments and suggestions!
> > >
> > > Warm regards,
> > >
> > > The Authors

---

### Official Review · Reviewer_ZDrZ · 2025-06-30

**Clarity:** 3
**Significance:** 3
**Originality:** 3
**Rating:** 4
**Confidence:** 3

**Summary:**

The paper introduces augmented in-context learning (ICL): a prompt that mixes a few labeled examples with many unlabeled inputs. Focusing on multi-class linear classification under an isotropic Gaussian-mixture model, the authors prove that a 4-layer transformer, when driven by chain-of-thought (CoT) prompting, can emulate an expectation–maximization (EM) algorithm. Experiments on synthetic 3-class, 3-dim Gaussian data confirm that accuracy improves consistently with more unlabeled points and surpasses the Bayes classifier trained on the labeled set alone.

**Questions:**

Please refer to the weaknesses section above.

**Ethical Concerns:**

["NO or VERY MINOR ethics concerns only"]

**Final Justification:**

The authors addressed most of my concerns. However, despite the additional experiments on AG News dataset, there is no comparison with existing baselines such as standard semi-supervised approached (e.g., pseudo-labelling). Its unclear how well their approach is suited for real world tasks.  Thus, I'll maintain my score at 4.

**Quality:**

3

**Strengths And Weaknesses:**

**Strengths**
- The core idea of augmenting a prompt with unlabeled data to improve In-context learning is interesting.
- The paper provides rigorous theory under multi-class linear classification setting for a multi-layer transformer
- A major strength of the work is the explicit construction of a 4-layer transformer that can emulate an Expectation-Maximization (EM) algorithm using chain-of-thought.
- The writing is very clear and easy to follow. Notation is kept minimal, derivations are step-by-step, and appendices contain complete proofs, making the work easy to follow and verify.

**Weaknesses**

- The 4-layer network is rigidly engineered: later layers are hard-wired to act only at specific CoT steps (e.g., Layer 4 fires solely on step 1 to initialize class means). The analysis confirms that if such a network is trained under teacher-forcing it will converge, but doesn't show that a standard autoregressive transformer would naturally converge to this this circuit.

- Theorem 4.2 requires the number of CoT steps (t) to grow with the number of unlabeled samples ($t \geq M^{1/4}$) to achieve the improved accuracy.  Since each CoT step is a forward pass over a sequence whose length also grows with $M$ and $t$, the total inference cost grows substantially with the amount of unlabeled data. The paper also doesn't provide any analysis of memory or latency, which is critical for practical tasks.

- All experiments use synthetic 3-dimensional, 3-class mixtures with just five labelled points. There are no language, vision, or large-scale tests, and no comparison against standard semi-supervised baselines (e.g., pseudo-labelling). Its unclear how well the theory translates to and how well this approach is suited for real world tasks.

- Figure 1 reports that the transformer exceeds the Bayes oracle trained on the labeled set for certain $\epsilon$ values, but that oracle ignores unlabeled data by construction. A fairer comparator would be an explicit EM or self-training classifier that does exploit the same unlabeled dataset.

- This work misses some key related works. For instance, the paper "Context is Environment (Gupta et al., ICLR 2024)" already provides theory and experiments showing that unlabeled context tokens improve ICL under distribution shift. The present manuscript neither cites nor contrasts this closely related work. Similarly as mentioned above, 'Many-Shot In-Context Learning (Agarwal et al.) also uses unlabelled queries to increase the context length and subsequently performance.

---

> ### Author Rebuttal · Authors · 2025-07-31
>
> We thank the reviewer for the careful reading and thoughtful comments. Please see our responses below.
>
> ***Weakness 1:*** The analysis doesn't show the convergence of the standard autoregressive transformer.
>
> ***Response to Weakness 1:*** We thank the reviewer's insightful comment. Indeed, our current theoretical construction employs a designed 4-layer transformer architecture, with specific layers activated only at certain CoT steps, to facilitate precise theoretical analysis and ensure provable convergence under teacher-forcing training.
>
> We note that explicitly constructing a transformer architecture to demonstrate specific capabilities during ICL is a common approach in the theoretical study of transformers [1–4], and our methodology follows this line of work. Building on this, we further show that such a transformer can be explicitly obtained through CoT training, with a rigorous analytical characterization of the non-asymptotic training dynamics.
>
> Analyzing the training dynamics of a general autoregressive transformer remains analytically intractable due to the inherent complexity and non-convexity of the activation functions. Even for single-layer transformers, tractable analysis often relies on simplified or specialized architectures. For instance, [5] investigates a single-layer linear transformer in the context of linear regression. Other works have similarly focused on linear transformers or restricted settings such as single-layer, single-head architectures with ReLU-activated attention for classification tasks [1,6], diverging from the standard autoregressive architecture.
>
> Moreover, we believe that starting with a general autoregressive transformer is not necessary to demonstrate our primary claim, i.e., unlabeled data can provably enhance in-context learning.
>
> For these reasons, in this work, we focus on a specifically constructed 4-layer transformer architecture. Extending our theoretical analysis to the standard autoregressive transformer is an important direction we leave for future work.
>
> ***Weakness 2:*** Need analysis of memory and latency.
>
> ***Response to Weakness 2:*** We thank the reviewer for this insightful comment. During inference with $M$ unlabeled samples, our method requires approximately $\mathcal{O}(M^{1/4})$ CoT steps.
> At any given step $t$, the input sequence length is on the order of $\mathcal{O}(N+M+M^{1/4})=\mathcal{O}(N+M)$. Since previous steps do not need to be stored, the peak memory requirement remains $\mathcal{O}(N+M)$.
>
> In terms of latency, the time complexity for a single forward pass under a standard transformer scales in $\mathcal{O}((N+M)^2)$ due to the self-attention mechanism. As established in Theorem 4.2, the time complexity under our method is $\mathcal{O}(N^2M^{1/4}+NM^{5/4}+M^{9/4})$. This is because under our method, the inference time for each forward pass in $\mathcal{O}((N+M+M^{1/4})^2)$, and we need in total $\mathcal{O}(M^{1/4})$ passes. When $M\gg N$, our time complexity becomes $\mathcal{O}(M^{9/4})$ and that of a single forward pass for a standard transformer is $\mathcal{O}(M^{2})$. We view this moderate polynomial overhead as a reasonable trade-off: Theorem 4.2 shows a non-asymptotic guarantee that the test error converges without bias. In contrast, running a standard transformer for one (or any fixed number of) passes leaves a persistent bias term in the accuracy, as shown in the concurrent work [7, Thm. 4].
>
> We thank the reviewer again for this valuable comment. We will incorporate this detailed complexity analysis into the revised manuscript.
>
> ***Weakness 3:*** It's unclear how well the theory translates to and how well this approach is suited for real-world tasks.
>
> ***Response to Weakness 3:*** We thank the reviewer for this important comment. We have performed an additional experiment on a real-world dataset to verify the impact of unlabeled samples on the ICL performance of a pre-trained LLM. Please refer to our responses to Reviewer RyNA (Response to Weakness 1) for the experimental setup and results.
>
> ***Weakness 4:*** Figure 1 missed the unlabeled classifier oracle.
>
> ***Response to Weakness 4:*** We thank the reviewer for this constructive feedback.
>
> Our primary theoretical contribution is to demonstrate that a transformer is capable of using unlabeled data to improve in-context learning. Our comparison to the standard Bayes oracle, which by definition cannot access unlabeled data, was intended to isolate and highlight this specific capability.
>
>
> We agree that including a semi-supervised baseline (such as an EM-based oracle) is a valuable addition to help us understand how well a transformer can utilize unlabeled data. We will add such baselines to our experiments in the revised manuscript to provide a more comprehensive evaluation.
>
>
>
> ***Weakness 5:*** Missing some related works.
>
> ***Response to Weakness 5:*** We thank the reviewer for pointing out these important related works.
>
> We note that Gupta et al. (2023) use unlabeled data in an **unsupervised** ICL setting, where the prompt only contains unlabeled data. In contrast, we utilize unlabeled data in addition to the labeled examples to augment ICL in a **semi-supervised learning** fashion. Our method also differs from Agarwal et al. (2024), which it proposes two methods for many-shot ICL, namely, 1) Reinforced ICL and 2) unsupervised ICL. For 1), it synthesizes new, fully-labeled examples by generating and selecting its own rationales. For 2), it simply inputs unlabeled demonstrations in the prompt. Both methods differ from our approach, where we mix both labeled and unlabeled examples in the prompt and theoretically analyze its performance.
>
> We will include and compare with those references and other relevant papers in the revision.
>
> References:
>
> [1] Ahn, Kwangjun, et al. "Transformers learn to implement preconditioned gradient descent for in-context learning." Advances in Neural Information Processing Systems 36 (2023)
>
> [2] Bai, Yu, et al. "Transformers as statisticians: Provable in-context learning with in-context algorithm selection." Advances in neural information processing systems 36 (2023)
>
> [3] Von Oswald, Johannes, et al. "Transformers learn in-context by gradient descent." International Conference on Machine Learning. PMLR, 2023.
>
> [4] Akyürek, Ekin, et al. "What learning algorithm is in-context learning? investigations with linear models." arXiv preprint arXiv:2211.15661 (2022).
>
> [5] Mahankali, Arvind, Tatsunori B. Hashimoto, and Tengyu Ma. "One step of gradient descent is provably the optimal in-context learner with one layer of linear self-attention." arXiv preprint arXiv:2307.03576 (2023).
>
> [6] Li, Hongkang, et al. "How do nonlinear transformers learn and generalize in in-context learning?." arXiv preprint arXiv:2402.15607 (2024).
>
> [7] Li, Yingcong, et al. "When and How Unlabeled Data Provably Improve In-Context Learning." arXiv preprint arXiv:2506.15329 (2025).
>
> ----------
>
> We hope our responses have addressed your concerns satisfactorily. We sincerely appreciate your constructive feedback and would be grateful if you would consider raising your evaluation based on our responses. We would also be happy to address any further questions you may have.

---

> > ### Comment · Reviewer_ZDrZ · 2025-08-04
> >
> > Thank you for your detailed response. The rebuttal has addressed most of my concerns and thus I will maintain my score at 4.

---

> > > ### Author Response · Authors · 2025-08-04
> > > **Thank you for your response**
> > >
> > > Dear reviewer ZDrZ:
> > >
> > > We are glad to know that our responses have addressed most of your concerns. We sincerely appreciate your helpful comments, positive feedback, and keeping your positive score. Thank you once again for your valuable input and for contributing to enhancing the quality of our work!
> > >
> > > Warm regards,
> > >
> > > The Authors

---

### Official Review · Reviewer_J6se · 2025-07-02

**Clarity:** 3
**Significance:** 3
**Originality:** 3
**Rating:** 5
**Confidence:** 1

**Summary:**

The paper presents some theoretical insights leveraging unlabeled data for ICL, in the multi-class linear classification setting. Key findings include that transformers can emulate EM-style algorithm with CoT prompting and such models can be trained using teacher forcing, resulting in provable ICL performance improvement. Empirical experiments are also conducted to validate the theoretical insights.

**Questions:**

1. \[Line 130\] It seems the normalization layers, which are commonly used, are missing here. Is there any justification?
2. \[Line 160\] The assumption that covariance is isotropic and shared across classes seems a strong one. How will the insights change if we relax these two assumptions?
3. Or if these two assumptions are essential for derivation, please elaborate in Section 3.2
4. \[Line 323\] The data dimension of 3 seems very small. Is there any justification for this? In practical settings, having tens or even hundreds of features seems typical.

**Ethical Concerns:**

["NO or VERY MINOR ethics concerns only"]

**Final Justification:**

The rebuttal has addressed my concerns, so I've raised my rating. However, my confidence remains low as I don't have much expertise in theoretical research

**Quality:**

3

**Strengths And Weaknesses:**

1. Unlike traditional semi-supervised settings, this paper explores leveraging unlabeled data directly, which is a new and interesting setting.
2. There has been previous work showing empirically that unlabeled data can boost ICL performance, even simply batching some queries (arxiv.org/abs/2405.09798). Providing theoretical insights is significant.
3. The implementation addressed in Section 5 is a good addition to the theoretical existence proof.

Weaknesses

1. The work is limited to the linear classification task, and there’s little discussion on the applicability of the insights beyond this setting.
2. The setup for empirical experiments (Section 6\) is not sufficiently justified. Many decisions seem important but not explained.

---

> ### Author Rebuttal · Authors · 2025-07-31
>
> We thank the reviewer for the careful reading and thoughtful comments. Please see our responses below.
>
> ***Weakness 1:*** The work is limited to the linear classification task, and there’s little discussion on the applicability of the insights beyond this setting.
>
> ***Response to Weakness 1:*** We appreciate the reviewer's valuable feedback. While our theoretical analysis indeed focuses on the multi-class linear classification task, this choice allows for rigorous analytical insights into the transformer's capability to utilize unlabeled data effectively. Importantly, our theoretical framework sheds light on the fundamental mechanism by which the transformer is able to leverage unlabeled data during ICL through CoT prompting. We believe our analysis can be extended beyond the linear classification task considered in this work.
>
> For example, a natural extension is to consider *generalized linear classification* tasks. In this framework, the label $ y \in \\{-1, 1\\}$ is modeled through a nonlinear decision boundary using a link function $\sigma: \mathbb{R} \to [0,1]$, such that the conditional probability of the label is given by
>
> $$
> \mathbb{P}(y = 1 \mid \mathbf{x}) = \sigma(\mathbf{w}^{*\top} \mathbf{x}),
> $$
>
> where  $\sigma$ is commonly chosen as the sigmoid function $ \sigma(z) = 1 / (1 + e^{-z})$, and $\mathbf{w}^* \in \mathbb{R}^d $ is an unknown parameter vector. This generalized linear model (GLM) setting introduces a richer hypothesis space while retaining mathematical tractability. In the context of augmented ICL, a key question is whether a transformer can utilize both the labeled and unlabeled examples to infer the latent structure governed by $\mathbf{w}^* $ and improve the ICL performance.
>
> We can construct a transformer and show that it can mimic a gradient descent step on the cross-entropy loss defined on the labeled examples plus an entropy regularization term defined on the unlabeled data. Such a regularization term can be used to improve the estimate of the optimal weights  $\mathbf{w}^*$ and prevent overfitting, especially when labeled data is scarce.
> Then, we can show that the transformer can be obtained through a CoT-based teacher-forcing training.  The training loss is the $\ell_2$ distance between the transformer's output at the $t$-th CoT step and the $t$-th gradient-descent step.
> We will explore such extensions as the next step of our research.
>
> ***Weakness 2:*** The setup for empirical experiments is not sufficiently justified.
>
> ***Weakness 2:***  We thank the reviewer for this helpful suggestion. The key parameter choices of the experiments are motivated as follows: The number of classes ($C=3$) and data dimension ($d=3$) are chosen to make the task more complicated than the binary case ($C=2$) while maintaining a low computational cost.
> Noise levels ($\epsilon \in \{0.7, 1.5\}$) are selected to assess our method's robustness across varying signal-to-noise ratios (SNRs). The choice of the two distinct levels allows us to test the model in both low-noise (high SNR) and high-noise (low SNR) regimes. Transformer input dimension ($d_p=16$) is a direct consequence of our proposed embedding architecture, as specified in Equation (3.2). This choice ensures consistency between our model's theoretical formulation and its practical implementation.
>
> We will integrate these justifications into the revised manuscript to improve clarity and reproducibility.
>
>
>
>
> ***Question 1:*** The commonly used normalization layers are not considered.
>
> ***Response to Question 1:***
> We thank the reviewer for highlighting this important point. Indeed, normalization layers are commonly utilized in transformer architectures. However, in our theoretical framework, we omit the normalization layers to simplify the analysis and provide clearer insights into the essential mechanics of the transformer's learning and inference processes. This choice follows recent theoretical ICL studies [1–4]. We will add this clarification about the normalization layers in our revised manuscript.
>
>
>
> ***Question 2:*** The assumption that covariance is isotropic and shared across classes seems a strong one. How will the insights change if we relax these two assumptions?
>
>
> ***Response to Question 2:*** We thank the reviewer for this insightful question. The assumption of a shared, isotropic covariance matrix is a strategic choice made for theoretical tractability. Specifically, this simplification ensures that the update rules within the attention block maintain an isotropic structure, which is crucial for deriving a tractable, closed-form solution.
>
> We note that this assumption is common and widely adopted in related literature to facilitate theoretical analysis [5, 6, 7].
>
> If we were to relax this condition, the model would need to estimate class-specific and anisotropic covariance structures. This alternative presents significant theoretical challenges, making the update rules difficult to analyze.
>
>
> ***Question 3:*** Need to elaborate if these two assumptions are essential.
>
> ***Response to Question 3:*** We thank the reviewer for the suggestion. We will revise Section 3.2 as follows (from line 176):
>
> *In this work, we assume $\mathbf{\Sigma}$ is isotropic. We adopt this assumption for theoretical tractability, as it is crucial for deriving the closed-form update rules for the transformer. This approach is a standard and widely adopted practice in related literature to facilitate theoretical analysis (He et al., 2025; Zhang et al., 2024; Chen et al., 2025).*
>
>
> ***Question 4:*** The data dimension is small.
>
> ***Response to Question 4:*** We thank the reviewer for this question. We choose $d=3$ to validate our theoretical results efficiently and with low computational cost. Importantly, our theoretical framework and conclusions are not limited to this specific dimensionality.
>
> We consider a more practical setting with much higher data dimension and conduct an additional experiment on a real-world dataset with a pre-trained LLM.  Please refer to our responses to Reviewer RyNA (Response to Weakness 1) for the experimental setup and results.
>
>
>
> References:
>
> [1] Uncovering mesa-optimization algorithms in transformers.
>
> [2] Transformers learn to implement pre-conditioned gradient descent for in-context learning.
>
> [3] Transformers as statisticians: Provable in-context learning with in-context algorithm selection.
>
> [4] When and How Unlabeled Data Provably Improves In-Context Learning
>
> [5] He et al., "Transformers versus the em algorithm in multi-class clustering." arXiv preprint arXiv:2502.06007 (2025).
>
> [6] Zhang et al., "In-context learning of a linear transformer block: Benefits of the MLP component and one-step DG initialization." Advances in Neural Information Processing Systems 37 (2024): 18310-18361.
>
> [7] Chen et al., "Transformers as Unsupervised Learning Algorithms: A study on Gaussian Mixtures." arXiv preprint arXiv:2505.11918 (2025).
>
>
> ----------
>
> We hope our responses have addressed your concerns satisfactorily. We sincerely appreciate your constructive feedback and would be grateful if you would consider raising your evaluation based on our responses. We would also be happy to address any further questions you may have.

---

### Official Review · Reviewer_RyNA · 2025-07-05

**Clarity:** 4
**Significance:** 4
**Originality:** 4
**Rating:** 5
**Confidence:** 2

**Summary:**

The paper introduces a new framework called augmented in-context learning (ICL) that leverages unlabeled data to improve the performance of transformers in classification task.
Augmented ICL Framework: The research proposes an augmented ICL framework where a transformer's prompt includes both a small set of labeled examples and a larger block of unlabeled inputs. This approach allows the transformer to infer missing labels within a single forward pass, bypassing the need for costly and time-consuming data labeling or pseudo-label generation.
Expressiveness with Chain-of-Thought (CoT) Prompting: The paper demonstrates that a multi-layer transformer, when prompted with Chain-of-Thought (CoT) reasoning, can effectively emulate an Expectation-Maximization (EM) algorithm for multi-class linear classification. This enables the transformer to extract useful information from both labeled and unlabeled data, leading to provable improvements in ICL accuracy. The study theoretically shows that the class mean estimation converges to the ground truth as CoT steps increase, and the approach's excess risk scales more favorably compared to methods using only labeled data.
Empirical Validation: Experiments show that the augmented ICL framework consistently outperforms conventional few-shot ICL in class mean estimation and label prediction, with the benefits becoming more significant as the amount of unlabeled data increases. Augmented ICL even surpasses the Bayes-optimal classifier that relies solely on labeled data.

**Questions:**

see

**Ethical Concerns:**

["NO or VERY MINOR ethics concerns only"]

**Limitations:**

Does language models trained on real-world data, where patterns are not as clear as in the synthetic data, align with the proposed theoretical explanation of unlabled ICL?

Does the proposed theory generalize to other architectures than transformer?

**Quality:**

4

**Strengths And Weaknesses:**

I am not familiar with this field. So my review will provide limited value.

## Strengths

This paper shows that transformer language model can learn from unlabeled ICL data from a theoretical perspective.

The paper is generally well-written and clear. The appendix provides comprehensive details about the proposed theory.

The paper's originality is high. The concept of integrating unlabeled data into ICL in a theoretically sound manner, specifically by showing how a transformer can emulate an EM algorithm through CoT prompting, is novel. While ICL and EM are not new, their combination in this specific, provable framework for transformers is a unique contribution that pushes the boundaries of our understanding of in-context learning.

## Weaknesses

Lacking experiments on real-world pre-training data.

---

> ### Author Rebuttal · Authors · 2025-07-31
>
> We thank the reviewer for the careful reading and thoughtful comments. Please see our responses below.
>
> ***Weakness 1:*** Lacking experiments on real-world pre-training data.
>
> ***Response to Weakness 1:***
> We thank the reviewer’s helpful comment. As a theoretical work, in our submission, we rely on a synthetic experiment to validate our theoretical claim, i.e., a transformer trained with standard teacher‑forcing can learn to exploit unlabeled data in addition to labeled examples to enhance its in‑context learning (ICL) accuracy.
>
>
> To validate the theoretical claim in more practical settings with a real-world dataset, as a first step, we conduct experiments on the AG News classification dataset with the Qwen3-8B model. The token embedding dimension is 4096, which is much higher than the dimension we consider in the synthetic experiment. We consider 4 candidate classes, and for each prompt, it consists of 4 labeled examples along with $1, 4$ or $8$ unlabeled samples. To encourage CoT reasoning, we use the following prompt:
>
>
> *First, think step-by-step, compare them with the patterns you saw
>  in both the labeled and unlabeled blocks, and decide the best category for each one.
>  You can compare with other unlabeled samples before making a final prediction.*
>
> For each prompt with $M$ unlabeled samples, we evaluate the model's performance after it completes its CoT reasoning process. We extract the final prediction for each of the $M$ samples from the model's generated output. These $M$ predictions are then compared against their ground-truth labels to calculate the overall prediction accuracy. The report results in the following table.
>
> | M | Test Accuracy |
> |-----|---------------|
> | 1   | 0.880 |
> | 4   | 0.896 |
> | 8   | 0.926 |
>
> We note that the average test accuracy, calculated over 100 prompts for each M, increases monotonically as M increases, which is consistent with our theoretical results.
>
> We will conduct more comprehensive experiments on a range of ICL tasks using real-world datasets and will report the results in the revised version.
>
>
> ***Limitation 1:*** Does language models trained on real-world data, where patterns are not as clear as in the synthetic data, align with the proposed theoretical explanation of unlabeled ICL?
>
> ***Response to Limitation 1:*** We thank the reviewer for the insightful question. As elaborated in our response to Weakness 1, we have performed an additional experiment on a real-world dataset to verify the impact of unlabeled samples on the ICL performance of a pre-trained LLM. We observe that although the assumptions we made for the theoretical analysis may not hold in this setting, the ICL performance monotonically improves as the number of unlabeled samples increases, which is aligned with our theoretical explanation of the augmented ICL.
> We will conduct more comprehensive experiments on a range of ICL tasks using real-world datasets to verify this and will report the results in the revised version.
>
> ***Limitation 2:*** Does the proposed theory generalize to other architectures than the transformer?
>
> ***Response to Limitation 2:*** We thank the reviewer for this question. Our work is intentionally focused on the transformer architecture, as ICL is a unique capability that is closely associated with and demonstrated by this class of models. Therefore, it is not straightforward to extend our theory to other deep learning architectures.
>
> However, we emphasize that our theory is not limited to the specific multi-layer transformer with CoT reasoning analyzed in the paper. It is generalizable across the transformer family. For example, our framework can be extended to a recursive transformer (or looped transformer). This would involve reformulating the explicit reasoning steps of CoT into implicit reasoning steps handled by the model's recurrent structure, which can be achieved by designing a suitable embedding scheme.
>
>
> ----------
>
> We hope our responses have addressed your concerns satisfactorily. We sincerely appreciate your constructive feedback and would be grateful if you would consider raising your evaluation based on our responses. We would also be happy to address any further questions you may have.

---

> > ### Comment · Reviewer_RyNA · 2025-08-06
> >
> > Thanks for your response. I will keep my score.

---

> ### Author Response · Authors · 2025-08-06
>
> Dear Reviewer RyNA:
>
> Thank you very much for going through our response and keeping your favorable rating! Your thoughtful comments have helped us greatly improve the quality of this work, and we will definitely keep polishing the paper and incorporating your valuable suggestions into the revision.
>
> Warm regards,
>
> The Authors

---

### Note · Authors · 2025-08-13

We thank the ACs and reviewers for their thoughtful engagement. There is broad agreement on the paper’s contributions:

- RyNA: “The paper's originality is high. The concept of integrating unlabeled data into ICL in a theoretically sound manner, specifically by showing how a transformer can emulate an EM algorithm through CoT prompting, is novel. While ICL and EM are not new, their combination in this specific, provable framework for transformers is a unique contribution that pushes the boundaries of our understanding of in-context learning.”


- J6se: “Unlike traditional semi-supervised settings, this paper explores leveraging unlabeled data directly, which is a new and interesting setting. ... Providing theoretical insights is significant.”


- ZDrZ: “The core idea of augmenting a prompt with unlabeled data to improve In-context learning is interesting.
The paper provides rigorous theory under multi-class linear classification setting for a multi-layer transformer
A major strength of the work is the explicit construction of a 4-layer transformer that can emulate an Expectation-Maximization (EM) algorithm using chain-of-thought.”

- uxcu: “It proposes a novel augmented ICL framework (apart from the concurrent work).
It is the first theoretical study showing the benefit of unlabeled data for ICL in transformers.”

Reviewers raised two primary concerns: (i) incomplete discussion of related work and missing citations, and (ii) the need to validate the theoretical results in a real-world setting. We addressed both.
First, we substantially revised the *Introduction* and *Related Work* sections to incorporate all references mentioned by the reviewers, along with additional relevant literature. Second, we added an experiment on the AG News dataset using the Qwen3-8B model to evaluate the impact of unlabeled examples on a pre-trained LLM’s ICL performance. The average test accuracy increases monotonically with the number of unlabeled samples, consistent with our theoretical predictions. We will include more comprehensive experiment results in our revised version.

Following these additions, reviewers indicated they were satisfied with our responses and raised no further concerns.
We believe these additions can be incorporated without requiring major changes to our original submission. We appreciate the reviewers’ feedback and thank them for helping us improve the paper.

---

### Decision · Program_Chairs · 2025-09-17

**Decision:**

Accept (poster)

**Comment:**

This paper proposes a novel augmented ICL framework that integrates labeled examples and unlabeled data into prompts, demonstrating theoretically and empirically that transformers can emulate the EM algorithm via CoT prompting to enhance ICL performance in multi-class linear classification. This is the first theoretical study on unlabeled data’s impact on transformer-based ICL.

Although the reviewers with highest ratings have the lowest confidence scores, all reviewers give positive final ratings. After reading the paper, reviews and discussions, I recognize the contribution and technical soundness of this paper. During rebuttal, the authors added experiments on real-world dataset with Qwen3-8B, clarified extensions to generalized linear models and experimental parameter choices, added complexity analysis to justify overhead. Given the theoretical novelty, rigorous analysis, and empirical validation (including real-world data), along with the authors’ efforts to address the reviewers' concerns, the paper is recommended for acceptance.